# FTSCommDetector: Discovering Behavioral Communities through Temporal Synchronization

## Abstract

Why do trillion-dollar tech giants AAPL and MSFT diverge into different response patterns during market disruptions despite identical sector classifications? This paradox (Figure 1) reveals a fundamental limitation: traditional community detection methods fail to capture synchronization-desynchronization patterns where entities move independently yet align during critical moments. To this end, we introduce FTSCommDetector, implementing our Temporal Coherence Architecture (TCA) to discover similar and dissimilar communities in continuous multivariate time series. Unlike existing methods that process each timestamp independently, causing unstable community assignments and missing evolving relationships, our approach maintains coherence through dual-scale encoding and static topology with dynamic attention. Furthermore, we establish information-theoretic foundations demonstrating how scale separation maximizes complementary information and introduce Normalized Temporal Profiles (NTP) for scale-invariant evaluation. As a result, FTSCommDetector achieves consistent improvements across four diverse financial markets (SP100, SP500, SP1000, Nikkei 225), with gains ranging from 3.5% to 11.1% over the strongest baselines. The method demonstrates remarkable robustness with only 2% performance variation across window sizes from 60 to 120 days, making dataset-specific tuning unnecessary, providing practical insights for portfolio construction and risk management.

## 1 Introduction

Human cognition demands categories, yet reality offers only behaviors. This fundamental tension between our need to classify and nature's tendency to evolve manifests starkly in financial markets. Consider tech titans AAPL and MSFT: both trillion-dollar Information Technology leaders, both occupying identical positions in every traditional taxonomy. Yet during the January 2025 AI market turbulence, they diverged into entirely different behavioral clusters, revealing how static classifications fail to capture the living dynamics of complex systems. Traditional methods freeze relationships in time, but actual systems evolve continuously: assets form temporary alliances during crises, sectors synchronize during volatility spikes, and communities reshape with economic trends.

This behavioral similarity paradigm revolutionizes portfolio construction and risk management. Sadly, naive sector classifications fundamentally mislead: despite identical GICS sector classifications as Information Technology giants, AAPL and MSFT exhibit entirely different behavioral patterns after the turbulence caused by tariffs and AI panic from February to April in 2025 (Figure 1).

Beyond these predetermined labels, existing clustering methods fail to handle continuous multivariate time series with complex temporal dependencies (Table 1). Static methods (DAEGC (Wang et al., 2019), GUCD (He et al., 2020), VGAER (Qiu et al., 2022)) treat timesteps as isolated snapshots, missing temporal continuity (Figure 1). Temporal methods designed for discrete events (JODIE (Kumar et al., 2019), TGN (Rossi et al., 2020)) are fundamentally incompatible with continuous observations where all entities evolve simultaneously. Recent approaches either focus on different tasks (ViTST (Li et al., 2023) for classification, Koopa (Liu et al., 2023) for forecasting) or lack essential components for temporal clustering (Deep Temporal Graph Clustering (Liu et al., 2024), Persistent Community Detection (Li et al., 2018) lack multi-scale encoders). Even sliding window approaches

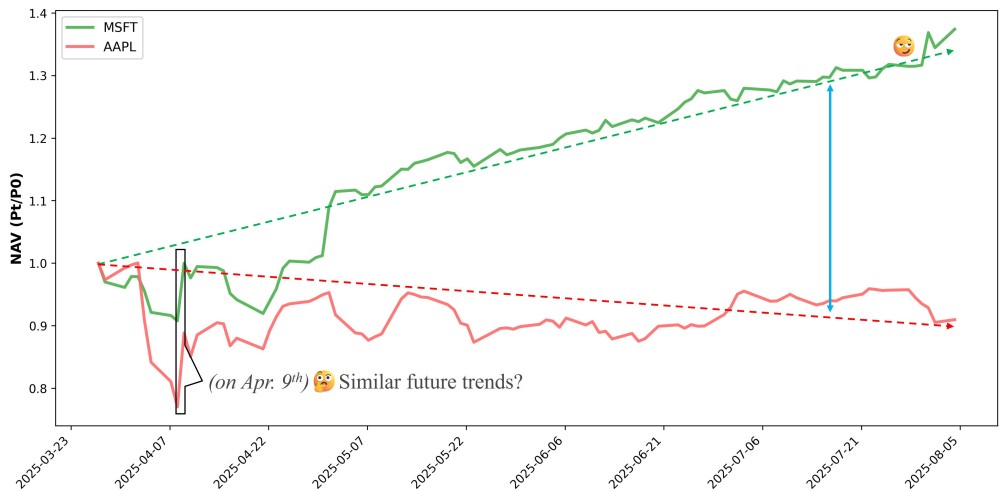

Figure 1: Temporal behavioral divergence exposing failures of snapshot clustering and sector grouping

process windows independently without shared learning, causing instability and missing recurring patterns, while frequent re-clustering for adapting to market changes incurs substantial transaction costs from portfolio rebalancing. Single-signal approaches using only price ratios overlook volume dynamics, volatility patterns, and cross-sectional dependencies that collectively determine behavioral communities. These limitations explain why existing methods failed to detect MSFT's alignment with AI growth stocks while AAPL clustered with stable mega-caps after the turbulence, since they cannot capture the multi-scale temporal dynamics that reveal true behavioral communities.

Therefore, our key insight here is: True communities are defined not by human-determined sector labels or snapshot-based similarities, but by synchronization, which means entities that move independently yet align during critical moments. This synchronization-desynchronization duality, overlooked by existing methods, forms the foundation of our Temporal Coherence Architecture (TCA).

We introduce *FTSCommDetector*, implementing TCA to address three fundamental challenges. First, **against temporal fragmentation** where existing methods process timestamps independently causing unstable assignments, we employ recurrent processing with shared parameters to ensure consistent pattern recognition. Second, **against single-scale myopia** that misses both short-term volatility and long-term trends, our hierarchical multi-scale modeling captures market dynamics at multiple frequencies simultaneously, distinguishing temporary fluctuations from persistent regime changes. Third, **against static rigidity** where traditional graphs cannot adapt to evolving relationships, our attention mechanisms dynamically reweight connections based on temporal context while maintaining stable topology, which can avoid the computational overhead of continuous graph reconstruction.

Our contributions include:

▷ **Conceptual Innovation:** We address the synchronization-desynchronization challenge through Temporal Coherence Architecture (TCA), which maintains stable communities while capturing dynamic behaviors. Unlike methods that process timestamps independently, TCA discovers communities through entities that move independently yet align during critical moments.

▷ **Empirical Validation:** FTSCommDetector achieves average improvements of 0.032 in IntraCorr and 0.039 in InterDissim across four diverse markets (SP100, SP500, SP1000, Nikkei 225), with consistent 3.5%-11.1% gains over strongest baselines. Remarkably, the method maintains 98% performance stability across window sizes (60-120 days), eliminating the instability plaguing sliding window approaches and dataset-specific tuning requirements.

▷ **Theoretical Foundations:** We establish information-theoretic principles demonstrating how multi-scale separation maximizes complementary information capture, introduce Normalized Temporal Profiles (NTP) for scale-invariant evaluation addressing market volatility variations, and provide generalization bounds connecting our approach to time-varying spectral clustering.

| Method | Continuous TS | Dynamic Attention | Multi-Scale |
|---|---|---|---|
| DAEGC (Wang et al., 2019) | ✗ | ✗ | ✗ |
| GUCD (He et al., 2020) | ✗ | ✗ | ✗ |
| VGAER (Qiu et al., 2022) | ✗ | ✗ | ✗ |
| CCGC (Yang et al., 2023) | ✗ | ✗ | ✗ |
| DGCLUSTER (Bhowmick et al., 2024) | ✗ | ✗ | ✗ |
| TGN (Rossi et al., 2020) | ✗ (events) | ✓ | ✗ |
| JODIE (Kumar et al., 2019) | ✗ (events) | ✗ | ✗ |
| DyGFormer (Yu et al., 2023) | ✗ (events) | ✓ | ✗ |
| **Ours** | ✓ | ✓ | ✓ |

Table 1: Comparison of key capabilities across community detection methods.

## 2 RELATED WORK

### 2.1 GRAPH COMMUNITY DETECTION WITH DEEP LEARNING

Static graph community detection has evolved through three main approaches. Attention-based methods pioneered by DAEGC (Wang et al., 2019) use attention mechanisms to capture node importance, with recent extensions like CoBFormer (Xing et al., 2024) addressing over-globalizing through bi-level transformers. Generative approaches including VGAER (Qiu et al., 2022) and GUCD (He et al., 2020) leverage variational inference and MRF-GCN respectively, focusing on reconstructing graph structures to discover communities: VGAER's modularity-based edge features particularly inspire our Net Asset Value (Pt/P0, NAV) based temporal contexts. Contrastive and geometric methods represent the latest 2023-2024 advances: CCGC (Yang et al., 2023) employs cluster-guided contrastive learning, DGCLUSTER (Bhowmick et al., 2024) performs modularity maximization, R-Clustering (Marco-Blanco & Cuevas, 2024) leverages random convolutional kernels, and LSEnet (Sun et al., 2024) exploits hyperbolic geometry for hierarchical structures. Despite these advances, all approaches process static snapshots independently, fundamentally missing the behavioral trajectories that define temporal communities.

### 2.2 TEMPORAL GRAPH LEARNING AND DYNAMIC COMMUNITY DETECTION

Temporal methods diverge based on their data assumptions. Discrete event models like TGN (Rossi et al., 2020) and JODIE (Kumar et al., 2019) maintain memory states for sporadic interactions (user clicks, messages), fundamentally incompatible with continuous multivariate observations where all entities evolve simultaneously. Parameter evolution approaches such as EvolveGCN (Pareja et al., 2020) adapt network parameters over time but still assume discrete snapshots, while transformer-based methods DyGFormer (Yu et al., 2023) and ROLAND (You et al., 2022) improve scalability through hierarchical updates. Continuous observation methods attempt to handle streaming data: Graph-based Hierarchical Clustering (Cini et al., 2023) and DSI (Zhang et al., 2024b) address temporal patterns but lack the multi-scale encoding essential for capturing volatility regimes, while recent time series transformers (Wang et al., 2024a; Zhang et al., 2024a) show promise for temporal modeling but focus on forecasting rather than community detection. None integrate the synchronization-desynchronization dynamics central to behavioral community discovery.

## 3 METHODOLOGY

### 3.1 OVERVIEW

We propose FTSCommDetector, a unified framework for community detection in multivariate time series that implements our Temporal Coherence Architecture (TCA) concept. As shown in Figure 2, FTSCommDetector integrates multi-scale temporal encoding, adaptive graph learning, and gated fusion enhanced by Set Transformers (Lee et al., 2019) for efficient processing. The TCA philosophy operationalizes temporal coherence through three principles: (1) Scale-Adaptive Encoding for capturing multi-granularity patterns, (2) Dynamic Connectivity Modeling maintaining

structural stability with temporal adaptivity, and (3) Multi-Stream Fusion synergistically combining graph and temporal representations. These principles guide the architectural design while the actual implementation focuses on computational efficiency and scalability.

## 3.2 PROBLEM STATEMENT

Given multivariate time series $\mathcal{X} = \{X_t\}_{t=1}^{T_{total}}$ where $X_t \in \mathbb{R}^{N \times D}$ represents $N$ entities with $D$ features, we construct sliding windows $\mathcal{W}_i$ of length $T$. With a static relationship graph $\mathcal{G} = (V, E, A)$, we learn $f : (\mathcal{W}_i, \mathcal{G}) \rightarrow \{C_1^i, \ldots, C_K^i\}$ identifying $K$ communities exhibiting similar temporal patterns and structural proximity. (Appendix A.2).

## 3.3 SCALE-ADAPTIVE ENCODING

The Scale-Adaptive Encoding layer captures temporal patterns at different granularities, recognizing that time series exhibit both short-term fluctuations and long-term trends equally important for community formation.

Our scale-adaptive design employs complementary temporal scales to maximize multi-scale representation power. The short-term pathway captures immediate dynamics and rapid fluctuations, while the long-term pathway spans extended periods encompassing gradual trends and structural shifts. This separation provides comprehensive temporal coverage without redundancy, essential for detecting communities that evolve at different timescales.

For input $X \in \mathbb{R}^{N \times D \times T}$, we process through two parallel pathways:

$$Z_{short} = \text{ShortTermEncoder}(X) \in \mathbb{R}^{N \times d_l \times T_s} \tag{1}$$

$$Z_{long} = \text{LongTermEncoder}(X) \in \mathbb{R}^{N \times d_g \times T_l} \tag{2}$$

where $d_l$ and $d_g$ are the short-term and long-term encoder dimensions, and $T_s$, $T_l$ are the reduced temporal dimensions after convolutional processing. Each encoder incorporates dual-attention mechanisms with channel and temporal modulation, achieving $O(CT)$ complexity compared to $O(T^2)$ for self-attention while preserving cross-dimensional dependencies crucial for multivariate patterns. Multi-scale temporal encoding with carefully separated receptive fields maximizes complementary information extraction through minimal feature overlap (Appendix A.5).

## 3.4 DYNAMIC CONNECTIVITY MODELING WITH NAV EDGE EMBEDDING

The Dynamic Connectivity Modeling layer employs static topology with temporally-adaptive attention weights enhanced by NAV-based edge features. Financial markets exhibit non-stationary dynamics with volatility clustering (Mandelbrot, 1963) and regime shifts (Hamilton, 1989), while fundamental relationships (sector membership, market capitalization) remain stable. We leverage this property through static graph structure with dynamic attention weights that capture time-varying interaction strengths.

We incorporate NAV-based modularity matrices $B^{NAV} = A^{NAV} - K$ (Appendix A.8) as edge features, where $A^{NAV}$ is the NAV correlation matrix and $K$ represents expected connections, providing implicit community supervision without the computational overhead of continuously reconstructing graph structures (Appendix A.8, B.3).

Given graph $\mathcal{G} = (V, E, A)$ constructed from temporal correlations (Appendix A.6) and temporal features $X \in \mathbb{R}^{N \times D \times T}$, we process temporal dynamics through a dual-BiLSTM architecture. First, we capture initial temporal patterns:

$$H^{(1)} = \text{BiLSTM}_1(X) \in \mathbb{R}^{T \times N \times h} \tag{3}$$

Next, a dynamic dependency module models evolving node interactions through adaptive graph propagation:

$$D^{(t)} = \text{ReLU}(\tanh(E_{node} \odot f(H^{(1,t)})) \cdot \tanh(E_{node} \odot f(H^{(1,t)}))^T) \tag{4}$$

where $E$ are learnable node embeddings and $f$ is a learned filter function (Appendix A.11).

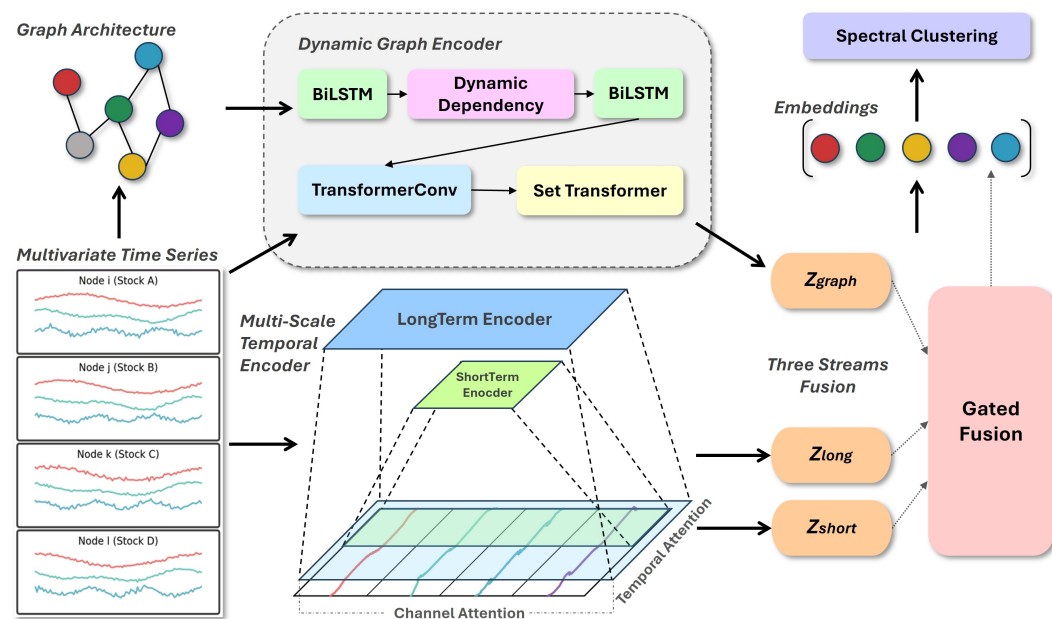

Figure 2: FTSCommDetector architecture: multi-scale temporal encoding, adaptive graph learning, and gated fusion

A second BiLSTM refines the representations, incorporating the dynamic dependencies:

$$H^{(2)} = \text{BiLSTM}_2(H^{(1)} + D^{(t)} \cdot H^{(1)}) \tag{5}$$

We then implement time-conditioned attention through temporal modulation. The hidden states are first modulated with temporal information:

$$\tilde{H}_i^{(t)} = H_i^{(2,t)} \odot \sigma(W_g \cdot \text{time}_t) + W_b \cdot \text{time}_t \tag{6}$$

where $\sigma$ denotes sigmoid gating and $\text{time}_t$ are learned time embeddings.

Query, Key, and Value computations incorporate both the modulated states and temporal context:

$$Q_i^{(t)} = W_Q \cdot \tilde{H}_i^{(t)} \tag{7}$$

$$K_i^{(t)} = W_K \cdot H_i^{(2,t)} + W_k \cdot \text{time}_t \tag{8}$$

$$V_i^{(t)} = W_V \cdot H_i^{(2,t)} + W_v \cdot \text{time}_t \tag{9}$$

Finally, attention scores produce the graph embedding:

$$\alpha_{ij}^{(t)} = \text{softmax}\left(\frac{Q_i^{(t)} \cdot K_j^{(t)}}{\sqrt{d_k}}\right), \quad Z_{graph} = \sum_{j \in \mathcal{N}(i)} \alpha_{ij}^{(t)} V_j^{(t)} \tag{10}$$

This formulation enables fully dynamic attention where Query modulation captures time-varying node states, while Key and Value updates reflect evolving relationships (Appendix A.12).

Bidirectional recurrent processing captures forward and backward temporal dependencies with $O(T)$ memory complexity compared to $O(T^2)$ for self-attention, crucial for long sequences (Appendix A.22). Set Transformers (Lee et al., 2019) achieve $O(NKT)$ (Appendix A.2,A.17) complexity for efficient aggregation. This architecture provides memory efficiency and training stability by keeping graph structure in memory once and allowing attention weights to evolve, which is particularly effective when connectivity patterns exhibit temporal persistence (Appendix A.7,A.9).

### 3.5 MULTI-STREAM FUSION

The multi-stream fusion layer integrates graph embeddings with multi-scale temporal features to create a comprehensive representation. This fusion is essential for capturing both structural relationships in the graph domain and dynamic patterns in the temporal domain. Gated fusion adaptively weights the information streams:

$$Z_{final} = \text{GatedFusion}([Z_{graph}, Z_{short}, Z_{long}]) \tag{11}$$

where the gated fusion applies learned gates to balance contributions from each stream based on the input characteristics. This design ensures that the model can dynamically emphasize either graph structure or temporal dynamics depending on the data patterns (Appendix A.10).

The architectural choices above are grounded in three key theoretical results that guide our design. First, the scale separation between short-term and long-term pathways is optimized to yield maximum complementary information for community detection through rate-distortion optimal encoding (Theorem 1, Appendix B.1). Second, our use of Normalized Temporal Profiles (NTP) ensures that discovered communities maximize behavioral coherence while remaining scale-invariant: the universal normalization $\text{NTP}_i(t) = X_i(t)/X_i(t_0)$ eliminates magnitude differences to reveal pure behavioral patterns across diverse financial markets (Theorem 2, Appendix B.2). Third, the multi-scale architecture provides implicit regularization: with probability $\geq 1 - \delta$, our clustering error satisfies $\mathcal{R}(h) \leq \hat{\mathcal{R}}(h) + O(\sqrt{K_{comm} \log(NT)/m})$ where $K_{comm}$ is the number of communities and $m$ is the number of training samples (Theorem 3, Appendix B.4).

### 3.6 ARCHITECTURE AND OPTIMIZATION

Three information streams fuse through a gated fusion network (Appendix A.10):

$$Z_{final} = \text{GatedFusion}(Z_{concat}) = \text{MLP}(Z_{concat}) \odot \sigma(W_g Z_{concat}) + W_r Z_{concat} \tag{12}$$

where $Z_{concat} = [Z_{graph}; Z_{short}.\text{flatten}(); Z_{long}.\text{flatten}()]$ concatenates all streams. The gating mechanism $\sigma(W_g Z_{concat})$ enables adaptive feature selection across different temporal scales, allowing the model to dynamically weight contributions based on context-specific importance. This design is crucial for capturing both structural graph patterns and multi-scale temporal dynamics.

The framework jointly optimizes reconstruction losses:

$$\mathcal{L}_{total} = \lambda_{graph} \mathcal{L}_{graph} + \lambda_{temporal} \mathcal{L}_{temporal} \tag{13}$$

where $\mathcal{L}_{graph}$ is the graph reconstruction loss and $\mathcal{L}_{temporal}$ is the multi-scale temporal reconstruction loss. These reconstruction objectives prevent representation collapse while maintaining computational efficiency. Loss weights are determined through validation (Appendix A.21, A.16).

## 4 EXPERIMENTS

### 4.1 EXPERIMENTAL SETUP

We conduct comprehensive experiments across diverse financial markets to evaluate the effectiveness of our temporal community detection framework. We utilize four datasets: SP100, SP500, SP1000, and Nikkei 225, representing different market segments and capitalization ranges (Appendix A.4). Implementation details are provided in Appendix C.2 and A.1.

### 4.2 BASELINES AND METRICS

**Baselines.** We compare FTSCommDetector against representative community detection methods spanning from established to recent approaches: DAEGC (Wang et al., 2019), GUCD (He et al., 2020), VGAER (Qiu et al., 2022), SDCN (Bo et al., 2020), CCGC (Yang et al., 2023), DGCLUS-TER (Bhowmick et al., 2024), and APDCG (Wang et al., 2024b). All baselines employ our identical dual-scale temporal encoder (TE), using the same multi-scale convolutional architecture adapted to their static graph constraints, isolating the comparison to graph learning mechanisms while providing equally powerful temporal features (Appendix A.18).

| Method | SP100 | | Nikkei225 | |
| --- | --- | --- | --- | --- |
| | IntraCorr | InterDissim | IntraCorr | InterDissim |
| DAEGC (Wang et al., 2019) + TE | 0.421±0.024 | 0.873±0.031 | 0.398±0.022 | 0.812±0.028 |
| GUCD (He et al., 2020) + TE | 0.438±0.015 | 0.907±0.026 | 0.413±0.017 | 0.846±0.021 |
| VGAER (Qiu et al., 2022) + TE | 0.452±0.016 | 0.931±0.022 | 0.427±0.011 | 0.871±0.024 |
| SDCN (Bo et al., 2020) + TE | 0.429±0.011 | 0.862±0.027 | 0.404±0.014 | 0.829±0.025 |
| CCGC (Yang et al., 2023) + TE | 0.471±0.013 | 0.956±0.016 | 0.448±0.010 | 0.912±0.018 |
| DGCLUSTER (Bhowmick et al., 2024) + TE | 0.487±0.009 | 0.978±0.014 | 0.463±0.008 | 0.894±0.020 |
| APDCG (Wang et al., 2024b) + TE | 0.479±0.007 | 0.993±0.023 | 0.456±0.015 | 0.927±0.017 |
| **Ours** | **0.504±0.012** | **1.016±0.019** | **0.496±0.005** | **0.938±0.011** |
| | SP500 | | SP1000 | |
| | IntraCorr | InterDissim | IntraCorr | InterDissim |
| DAEGC (Wang et al., 2019) + TE | 0.386±0.018 | 0.751±0.025 | 0.347±0.019 | 0.689±0.029 |
| GUCD (He et al., 2020) + TE | 0.402±0.016 | 0.783±0.027 | 0.368±0.014 | 0.724±0.026 |
| VGAER (Qiu et al., 2022) + TE | 0.417±0.014 | 0.806±0.023 | 0.382±0.017 | 0.751±0.020 |
| SDCN (Bo et al., 2020) + TE | 0.394±0.021 | 0.768±0.028 | 0.356±0.012 | 0.706±0.024 |
| CCGC (Yang et al., 2023) + TE | 0.439±0.011 | 0.842±0.021 | 0.409±0.016 | 0.827±0.020 |
| DGCLUSTER (Bhowmick et al., 2024) + TE | 0.451±0.010 | 0.871±0.012 | 0.401±0.009 | 0.798±0.016 |
| APDCG (Wang et al., 2024b) + TE | 0.457±0.015 | 0.859±0.019 | 0.416±0.010 | 0.813±0.012 |
| **Ours** | **0.490±0.013** | **0.926±0.017** | **0.462±0.015** | **0.892±0.021** |

Table 2: Performance comparison on temporal community detection across financial markets.[*]
[*]TE denotes our temporal encoder adapted for baselines: the same dual-scale architecture used in FTSCommDetector but constrained to static graphs per baseline requirements.

**Evaluation Strategy.** We employ NAV-based evaluation throughout training and validation to ensure consistency and avoid circular validation. NAV for entity $i$ is defined as $\text{NAV}_i(t) = P_i(t)/P_i(t_0)$, capturing relative price movements independent of absolute values. During training, early stopping uses NAV clustering quality metrics computed directly from price data, evaluating intra-cluster correlation and inter-cluster dissimilarity through a weighted composite score $S = w_{intra} \cdot S_{intra} + w_{inter} \cdot S_{inter}$, where asymmetric weighting favors discovering distinct market regimes over enforcing artificial homogeneity within clusters (Appendix A.13).

For final evaluation, following work on financial time-series clustering (Álvaro Cartea et al., 2023; Tola et al., 2008; de Prado, 2016), we compute comprehensive correlation-based metrics from ground-truth price data. Let $X_i \in \mathbb{R}^{T \times D}$ denote the feature matrix for entity $i$ over $T$ timesteps with $D$ features. For entities $i$ and $j$, the multi-feature correlation is computed as:

$$\rho_{ij} = \frac{1}{D} \sum_{d=1}^{D} |\text{Corr}(\tilde{X}_i^{(d)}, \tilde{X}_j^{(d)})| \tag{14}$$

where $\tilde{X}_i^{(d)} = (X_i^{(d)} - \mu_i^{(d)})/\sigma_i^{(d)}$ is the standardized $d$-th feature, with $\mu_i^{(d)}$ and $\sigma_i^{(d)}$ being the mean and standard deviation of feature $d$ for entity $i$. The absolute value ensures $\rho_{ij} \in [0, 1]$, where values close to 1 indicate strong correlation (either positive or negative).

Given clustering $\mathcal{C} = \{C_1, ..., C_K\}$, we define:

- *Intra-cluster Correlation*: $\text{IntraCorr} = \frac{1}{K} \sum_{k=1}^{K} \frac{2}{|C_k|(|C_k|-1)} \sum_{i<j \in C_k} \rho_{ij} \in [0, 1]$

- *Inter-cluster Correlation*: $\text{InterCorr} = \frac{2}{K(K-1)} \sum_{k<l} \frac{1}{|C_k||C_l|} \sum_{i \in C_k, j \in C_l} \rho_{ij} \in [0, 1]$

- *Inter-cluster Dissimilarity*: $\text{InterDissim} = 1 - \text{InterCorr} \in [0, 1]$

Higher values are better for both IntraCorr (indicating strong within-cluster correlation) and InterDissim (indicating weak between-cluster correlation). These metrics provide a comprehensive assessment of clustering quality, ensuring unbiased evaluation grounded in actual market behavior rather than learned embeddings (Appendix A.14).

| Mode | IntraCorr | InterDissim |
|---|---|---|
| Static | 0.468 ± 0.014 | 0.942 ± 0.026 |
| Basic | 0.481 ± 0.011 | 0.965 ± 0.024 |
| Enhanced | 0.493 ± 0.009 | 0.991 ± 0.016 |
| **Full** | **0.504 ± 0.012** | **1.016 ± 0.019** |

Table 3: Dynamic attention progression on SP100.

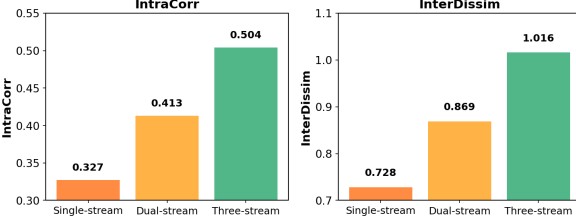

Figure 3: Multi-stream fusion ablation on SP100.

### 4.3 Main Results

Table 2 demonstrates that FTSCommDetector achieves competitive performance across diverse financial markets. On SP100, our method achieves 0.504 intra-cluster correlation and 1.016 inter-cluster dissimilarity, outperforming the strongest baseline by 0.017(3.5%) and 0.023 respectively (DGCLUSTER for IntraCorr, APDCG for InterDissim). The performance advantage is more pronounced against earlier methods, with improvements of 0.083 on intra-cluster correlation compared to DAEGC. FTSCommDetector achieves strong performance across all datasets. The improvements vary across markets, with gains of 0.033 on Nikkei225 IntraCorr, 0.033 on SP500 IntraCorr, and 0.046(11.1%)-0.065 on SP1000 metrics, reflecting different market characteristics and the challenges of consistent performance across diverse trading environments.

### 4.4 Application Case Studies

**Dynamic Market Regime Detection.** FTSCommDetector captures evolving market structures through behavioral clustering across 1000+ windows from 2020-2025. Figure 4 reveals the GameStop/Reddit event's impact (January-June 2021), where markets fragmented into 6 distinct communities: high-growth tech (AMD, TSLA) declined 21% while financials/energy (BAC, XOM) surged 38%, despite traditional sector classifications grouping them together. Additional market disruption events, including the January 2025 tech sector correction, are analyzed in Appendix A.27.

**Portfolio Construction Beyond Sectors.** FTSCommDetector uncovers behavioral correlations invisible to traditional sector classifications. During the GameStop event, airlines and hospitality stocks clustered together despite different GICS codes, while tech companies fragmented into three distinct behavioral groups. Our method maintains stability (averaging 2.3 clusters) with targeted sensitivity to market disruptions (Appendix Figure C.3), validated across both SP100 and Nikkei 225 markets (Appendix C.4). This enables dynamic portfolio strategies that exploit behavioral divergences from sector norms for enhanced indexing and statistical arbitrage.

### 4.5 Ablation Studies

**Dynamic Attention Progression.** We evaluate four temporal attention modes to validate our design choices (Table 3). *Static* uses fixed weights after threshold filtering. *Basic* adds weighted attention with static edge attributes. *Enhanced* incorporates multi-head attention with edge-modulated K and V. *Full* includes all dynamic components with temporal conditioning (Appendix A.23). The progression from Static (0.468 IntraCorr) to Full (0.504 IntraCorr) demonstrates consistent improvements, with the full model matching our reported performance.

**Kernel Size Ratio.** We analyze the impact of different kernel sizes for the dual-scale temporal encoders. While the architecture shows robustness across various ratios (performance within 5% variation), the 9:1 configuration provides marginal improvements through balanced multi-scale coverage.

**Window Size Robustness.** We evaluate window lengths from 60 to 120 days across multiple markets (Appendix A.22). FTSCommDetector demonstrates stability with approximately 2% performance variation across different window sizes, confirming its robustness to this hyperparameter. We adopt 89 days as it offers a practical balance between temporal coverage and computational efficiency.

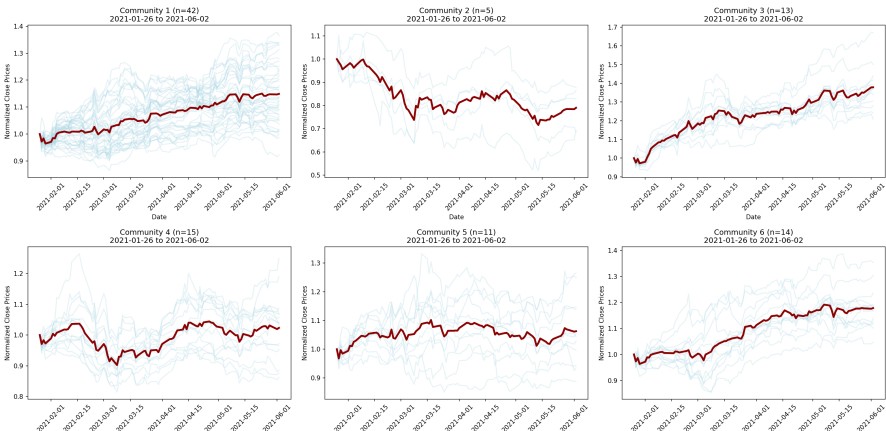

Figure 4: GameStop event (Jan-Jun 2021): Market fragmentation into 6 distinct behavioral clusters.

**Multi-Stream Fusion.** We compare three architectural variants: single-stream (graph only), dual-stream (graph + merged temporal), and three-stream (graph + short-term + long-term). As shown in Figure 3, the three-stream architecture substantially outperforms single-stream with gains of 0.177 in IntraCorr and 0.288 in InterDissim on SP100, with consistent improvements on other datasets (Appendix A.24).

## 5 FUTURE WORK

Four primary directions for extending FTSCommDetector. **Theoretical foundations:** Establishing formal connections between our NAV-based modularity embedding and spectral clustering theory will strengthen theoretical guarantees (Appendix A.20). **Ground truth consensus:** Our unsupervised approach discovers patterns without labels. Developing community-wide benchmarks with expert-validated ground truth for temporal clustering will provide more rigorous comparative evaluation and drive methodological advances. **Scale extensions:** Nyström approximation can reduce clustering complexity from $O(N^3)$ to $O(NK^2)$ for larger graphs. Graph sparsification can achieve $O(N \log N)$ construction. Knowledge distillation and pruning techniques support edge deployment. **Adaptive mechanisms:** Learning-based selection of window lengths, number of scales, and inducing points reduces tuning, improving generalization across domains. Additionally explore learnable topology evolution to capture structural changes beyond our current static topology design (Appendix A.19).

## 6 CONCLUSION

FTSCommDetector implements Temporal Coherence Architecture (TCA) to discover behavioral communities through synchronization patterns in multivariate time series. The dual-scale encoding captures short-term fluctuations and long-term trends while maintaining $O(NKT)$ complexity via Set Transformers. Static topology with dynamic features enables expressive temporal modeling without continuous graph reconstruction overhead.

Experiments across four financial markets demonstrate average improvements of 0.032 IntraCorr and 0.039 InterDissim, with gains from SP100 (+3.5%) to SP1000 (+11.1%) and Nikkei 225 (+7.1%). Performance varies only 2% across 60-120 day windows, ensuring robust generalization and direct applicability to diverse markets without reconfiguration, which is crucial for practical deployment.

The framework reveals hidden structures: airlines clustering with hospitality during crises, Japanese semiconductors aligning with domestic rather than global partners. These insights enable dynamic portfolio construction integrating behavioral patterns with sector knowledge. By unifying temporal dynamics with community detection through NAV-based evaluation and multi-scale encoding, FTSCommDetector establishes a robust paradigm for analyzing evolving relationships in continuous time series.

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

APPENDIX

# A  IMPLEMENTATION AND TECHNICAL DETAILS

## A.1  IMPLEMENTATION DETAILS

## A.2  VARIABLE DEFINITIONS AND NOTATION

Throughout this paper, we use the following consistent notation:

- $\mathcal{X} = \{X_t\}_{t=1}^{T_{total}}$: Complete time series dataset
- $\mathcal{W}_i$: $i$-th sliding window of the time series
- $\mathcal{G} = (V, E, A)$: Static relationship graph with vertices $V$, edges $E$, adjacency matrix $A$
- $N$: Number of entities (nodes) in the graph
- $T$: Temporal window length (Appendix A.22)
- $T_{total}$: Total length of the time series
- $D$: Number of input features per entity. In our experiments, $D = 7$ corresponding to:
    1. Daily log return
    2. Cumulative log return (1 week)
    3. Cumulative log return (2 weeks)
    4. Cumulative log return (1 month)
    5. Cumulative log return (2 months)
    6. Relative Strength Index (RSI)
    7. Moving Average Convergence Divergence (MACD)
- $d_{latent}$: Main latent dimension
- $h$: Hidden dimension for graph encoder
- $d_l = d_{latent}/2$: Short-term encoder latent dimension
- $d_g = d_{latent} - d_l = d_{latent}/2$: Long-term encoder latent dimension
- $T_s$: Reduced temporal dimension from short-term encoder (typically 8-9 after convolution)
- $T_l$: Reduced temporal dimension from long-term encoder (typically 5 after convolution)
- $K$: Number of inducing points in Set Transformer
- $K_{comm}$: Number of communities/clusters
- $C_k$: The $k$-th cluster/community
- $B^{NAV}$: NAV-based modularity matrix
- $k$: Convolutional kernel size (5 for short-term, 45 for long-term)
- $d_k$: Key dimension in attention mechanism
- $s$: Convolutional stride (3 for short-term, 11 for long-term)
- $r$: Channel attention reduction ratio
- $\lambda_{graph}, \lambda_{temporal}$: Loss component weights for graph and temporal reconstruction
- $m$: Number of training samples (windows)
- $\odot$: Element-wise (Hadamard) product
- $W_{ij}^{static}$: Static edge weights between entities $i$ and $j$
- $E_{node} \in \mathbb{R}^{N \times h}$: Learnable node embeddings
- $n_h$: Number of attention heads
- $W_g, W_b, W_k, W_v, W_r$: Learnable weight matrices for temporal modulation
- $W_O$: Output projection matrix for multi-head attention
- $W_Q, W_K, W_V \in \mathbb{R}^{d \times d}$: Query, Key, Value projection matrices for attention
- $Z_{graph}, Z_{short}, Z_{long}, Z_{final}$: Graph, short-term, long-term, and final embeddings

- $Z_{concat}$: Concatenated embeddings from all streams
- $\rho_{ij}$: Correlation coefficient between entities $i$ and $j$
- $\alpha_{ij}^{(t)}$: Attention weight from entity $i$ to $j$ at time $t$
- $\sigma$: Sigmoid activation function
- $\mu_i^{(d)}, \sigma_i^{(d)}$: Mean and standard deviation of feature $d$ for entity $i$
- $\mathcal{L}_{total}, \mathcal{L}_{graph}, \mathcal{L}_{temporal}$: Total, graph, and temporal loss functions
- $\mathcal{R}(h), \hat{\mathcal{R}}(h)$: True and empirical risk functions
- $\delta$: Confidence parameter for PAC bounds
- $\text{Corr}(\cdot, \cdot)$: Pearson correlation function
- $\text{NTP}_i(t) = X_i(t)/X_i(t_0)$: Normalized Temporal Profile
- $\tau = 0.75$: Correlation threshold for graph construction
- $k_l = 5, k_g = 45$: Short-term and long-term kernel sizes (used in theorems)
- $k_m$: Intermediate kernel size (used in Theorem 1)
- $\mathcal{N}(i)$: Neighborhood of node $i$ in the graph

### A.3 CORE ALGORITHM PSEUDOCODES

We present pseudocodes for our three core algorithmic contributions, emphasizing intuition over implementation details.

---

**Algorithm 1** Dual-Scale Temporal Encoder

---

**Require:** Time series $X \in \mathbb{R}^{N \times D \times T}$
**Ensure:** Multi-scale features $\{Z_{local}, Z_{global}\}$
1: **// Parallel multi-scale encoding**
2: $Z_{local} \leftarrow \text{ShortTermEncoder}(X)$ ▷ Short-term patterns
3:    **for** each Conv layer **do**
4:       Apply kernel=5, stride=3 ▷ Short-term patterns
5:       $H_{ch} \leftarrow \sigma(\text{FC}(\text{AvgPool}(H)) + \text{FC}(\text{MaxPool}(H)))$
6:       $H_{temp} \leftarrow \sigma(\text{Linear}(H.\text{mean}(\text{channels})))$
7:       $H \leftarrow H \odot H_{ch} \odot H_{temp}$ ▷ Dual attention
8:    **end for**
9: $Z_{global} \leftarrow \text{LongTermEncoder}(X)$ ▷ Long-term patterns
10:    Apply kernel=45, stride=11 ▷ Long-term trends
11:    Apply same attention mechanism
12: **// Adaptive fusion**
13: $w \leftarrow \text{Softmax}(\text{LearnableWeights})$
14: **return** $\{w_1 \cdot Z_{local}, w_2 \cdot Z_{global}\}$

---

**Algorithm 2** Graph Encoder with Temporal Aggregation

---

**Require:** Static graph $G = (V, E, A)$, Features $X \in \mathbb{R}^{N \times D \times T}$
**Ensure:** Graph embeddings $Z_{graph} \in \mathbb{R}^{N \times h}$
1: **// Temporal feature processing**
2: $H \leftarrow \text{BiLSTM}(X)$ ▷ Bidirectional temporal encoding
3: **// Efficient temporal aggregation**
4: $\mathcal{I} \in \mathbb{R}^{K \times h}$ ▷ Learnable inducing points, $K = 16$
5: $H_I \leftarrow \text{MultiheadAttn}(\mathcal{I}, H, H)$ ▷ Attend to temporal features
6: $Z_{agg} \leftarrow \text{PMA}(\text{ISAB}(H, H_I))$ ▷ Pool to single vector per node
7: **// Graph convolution with dynamic attention**
8: $Z_{graph} \leftarrow \text{TransformerConv}(Z_{agg}, E)$ ▷ Attention from node features (Shi et al., 2021)
9: **return** $Z_{graph}$

---

A.4 DATASET CHARACTERISTICS

Our experiments utilize four financial market datasets, each representing different market segments and characteristics:

**SP100:** Comprises the 100 largest U.S. companies by market capitalization from within the S&P 500 index. These blue-chip stocks provide stable trading patterns with high liquidity and extensive analyst coverage.

**SP500:** Encompasses 500 large-cap U.S. companies across all major sectors, accounting for approximately 80% of total U.S. market capitalization according to S&P Dow Jones Indices. This broader index captures more diverse market dynamics while maintaining focus on established companies.

**SP1000:** Combines the S&P MidCap 400 and S&P SmallCap 600 indices to form a benchmark for the mid-to-small cap segment of the U.S. equity market. These stocks exhibit higher volatility and less correlated movements, providing a challenging test case for community detection.

**Nikkei 225:** The premier index of the Tokyo Stock Exchange, comprising 225 large Japanese companies selected for liquidity and sector balance. This dataset enables cross-market validation and captures distinct Asian market dynamics.

All datasets span five years of daily trading data with consistent preprocessing. Price data and fundamental metrics were obtained from standard financial data providers, with SP100 data sourced from publicly available repositories and the remaining datasets obtained through proprietary financial data terminals under institutional access.

A.5 DUAL-SCALE TEMPORAL ENCODER: ARCHITECTURE OVERVIEW

Our dual-scale temporal encoder processes input at two complementary temporal resolutions to capture both short-term fluctuations and long-term trends. The architecture employs kernel sizes of 45 and 5 (9:1 receptive field ratio), selected to maximize complementary information based on our rate-distortion analysis while maintaining computational efficiency.

**ShortTermEncoder Architecture (Short-term patterns):**

- Kernel size: $k = 5$ (captures short-term local patterns)
- Stride: $s = 3$ (overlapping segments for fine-grained analysis)
- Architecture: Two-layer convolutional encoder with kernel size 5 and stride 3 for capturing local patterns
- Temporal progression: $T = 89 \rightarrow 29 \rightarrow 9$ (progressive compression)
- Decoder: Mirror architecture with ConvTranspose1d operations

**LongTermEncoder Architecture (Long-term trends):**

- Kernel size: $k = 45$ (captures bi-monthly/quarterly patterns)
- Stride: $s = 11$ (coarse-grained sampling for global trends)
- Architecture: Single-layer convolutional encoder with kernel size 45 and stride 11 for global patterns
- Temporal progression: $T = 89 \rightarrow 5$ (aggressive global compression)
- Decoder: ConvTranspose1d for reconstruction

**Design Rationale:** The complementary kernel sizes maximize multi-scale representation power through strategic temporal coverage. The local kernel with size 5 focuses on capturing short-term dynamics and microstructure patterns that emerge over daily to weekly timescales, essential for detecting rapid market reactions and transient volatility clusters. The global kernel spanning 45 timesteps encompasses longer periods that reveal structural shifts and regime changes occurring over monthly to quarterly horizons. This deliberate scale separation minimizes feature overlap while ensuring comprehensive temporal coverage: the local pathway extracts high-frequency signals that would be smoothed out by the global kernel, while the global pathway identifies persistent trends

invisible to the local kernel. Together, this multi-scale design enables the model to simultaneously capture both fine-grained patterns and long-term trends, providing superior temporal granularity compared to single-scale approaches that must compromise between resolution and context.

**Attention Mechanism:** For hidden representation $H \in \mathbb{R}^{B \times C \times T}$ at any layer:

$$\text{ChannelAttn}(H) = \sigma(\text{FC}(\text{AvgPool}(H)) + \text{FC}(\text{MaxPool}(H))) \tag{15}$$

$$\text{where FC}: \mathbb{R}^C \to \mathbb{R}^{C/r} \to \mathbb{R}^C, r = 16 \tag{16}$$

$$\text{TemporalAttn}(H) = \sigma(\text{Linear}(\text{GELU}(\text{Linear}(H.\text{mean}(\text{dim} = 1))))) \tag{17}$$

$$\text{DualAttn}(H) = H \odot \text{ChannelAttn}(H).\text{unsqueeze}(2) \odot \text{TemporalAttn}(H).\text{unsqueeze}(1) \tag{18}$$

The cross-attention design creates bidirectional information flow: channel attention identifies important features across the temporal dimension, and temporal attention weights time points based on feature relevance. This dual modulation creates a multiplicative interaction that captures complex spatio-temporal patterns.

### A.6 GRAPH CONSTRUCTION FROM TEMPORAL CORRELATIONS

Given temporal features $X \in \mathbb{R}^{N \times D \times T}$ within a sliding window, we first apply feature-wise normalization to ensure stability across different feature scales. Following the standardization approach recommended by LeCun et al. (LeCun et al., 1998), each feature dimension is normalized independently across all nodes and timesteps within the window:

$$\hat{X}_{:,d,:} = \frac{X_{:,d,:} - \mu_d}{\sigma_d + \epsilon} \tag{19}$$

where $\mu_d$ and $\sigma_d$ are the mean and standard deviation of feature $d$ computed across all nodes and timesteps, and $\epsilon = 10^{-8}$ for numerical stability. This feature-wise normalization ensures that features with different scales (e.g., returns vs. technical indicators) contribute equally to the correlation computation.

We then construct the graph topology through correlation-based adjacency with optional structural enhancements:

$$C_{ij} = \text{Corr}(X_i, X_j) \quad \text{(Pearson correlation across } T \text{ timesteps)} \tag{20}$$

$$A_{ij}^{base} = \begin{cases} 1 & \text{if } C_{ij} \geq \tau_{corr} \\ 0 & \text{otherwise} \end{cases} \tag{21}$$

$$A_{ij} = A_{ij}^{base} + \delta \cdot \mathbb{1}[\text{sector}_i = \text{sector}_j] \quad \text{(optional sector bonus)} \tag{22}$$

where $\tau_{corr} = 0.75$ is the correlation threshold and $\delta = 0.1$ is the sector bonus weight. These values were determined through extensive grid search on validation data to balance graph sparsity with sufficient connectivity for message passing.

**Fundamental Data and Sector Classification:** The correlation computation utilizes company fundamental data collected at the beginning of the 5-year data period, including financial metrics such as P/E ratios, market capitalization, beta, and other fundamental indicators. These static features provide a stable basis for structural relationship modeling. The sector classification for the optional sector bonus follows standard industry taxonomies: for SP100, SP500, and SP1000, we use GICS (Global Industry Classification Standard) sector codes; for Nikkei 225, we use TSE (Tokyo Stock Exchange) sector classifications. This sector information enhances graph connectivity between companies within the same industry, capturing domain knowledge about business relationships.

The resulting adjacency matrix $A$ encodes structural relationships that are fixed during processing as node features evolve dynamically, with noise robustness and computational efficiency through single-pass graph construction.

## A.7 Graph Encoder with Temporal Aggregation: Mathematical Formulation

**BiLSTM Temporal Processing:** Given input $X \in \mathbb{R}^{N \times D \times T}$, we process temporal sequences through bidirectional LSTMs:

$$h_0, c_0 = \mathbf{0} \in \mathbb{R}^{2 \times N \times h} \tag{23}$$

$$H, (h_T, c_T) = \text{BiLSTM}(X, (h_0, c_0)) \tag{24}$$

$$H = H^{\rightarrow} + H^{\leftarrow} \in \mathbb{R}^{T \times N \times h} \tag{25}$$

where $h$ is the hidden dimension. The BiLSTM captures bidirectional temporal dependencies, providing richer node representations.

**Set Transformer Aggregation:** We aggregate temporal features using Set Transformers with inducing points:

$$\mathcal{I} \in \mathbb{R}^{K \times h} \quad \text{(learnable inducing points, } K = 16) \tag{26}$$

$$H_I = \text{MultiheadAttn}(\mathcal{I}, H, H) \in \mathbb{R}^{K \times h} \tag{27}$$

$$H_{out} = \text{MultiheadAttn}(H, H_I, H_I) \in \mathbb{R}^{N \times T \times h} \tag{28}$$

$$S \in \mathbb{R}^{1 \times h} \quad \text{(learnable seed vector)} \tag{29}$$

$$Z_{agg} = \text{MultiheadAttn}(S, H_{out}, H_{out}) \in \mathbb{R}^{N \times h} \tag{30}$$

This achieves $O(NKT)$ complexity compared to $O(N^2 T^2)$ for full self-attention.

**Graph Convolution:** Inspired by TransformerConv (Shi et al., 2021), we compute multi-head attention over graph edges where attention weights are dynamically computed from the node features $h^{(t)}$ produced by BiLSTM, allowing edge importance to adapt to temporal context. Specifically, we use multi-head attention with $K_{heads}$ attention heads, where each head learns different aspects of node relationships. The neighbors $\mathcal{N}(i)$ for each node $i$ are determined by the graph structure, and the attention mechanism learns to weight these connections based on the current node states. This design exploits the observation that connectivity patterns exhibit temporal persistence, allowing efficient computation and stable training.

## A.8 NAV-Based Edge Embedding

**Motivation:** To implicitly optimize for community modularity, we incorporate NAV-based modularity matrices as edge features in the graph encoder. This provides direct supervision for community-aware graph learning without requiring explicit community labels.

**NAV Modularity Matrix:** For each temporal window with price data $P \in \mathbb{R}^{T \times N}$, we compute:

1. *NAV Correlation Matrix:* First, calculate NAV for each entity:

$$\text{NAV}_i(t) = \frac{P_i(t)}{P_i(t_0)} \tag{31}$$

Then compute the correlation matrix:

$$A_{ij}^{NAV} = \text{Corr}(\text{NAV}_i, \text{NAV}_j) \tag{32}$$

2. *Null Model:* The expected edge weight under random connections:

$$K_{ij} = \frac{d_i \cdot d_j}{2m} \tag{33}$$

where $d_i = \sum_j A_{ij}^{NAV}$ is the weighted degree and $m = \frac{1}{2} \sum_{ij} A_{ij}^{NAV}$ is the total edge weight.

3. *Modularity Matrix:* The difference between actual and expected connections:

$$B_{ij}^{NAV} = A_{ij}^{NAV} - K_{ij} \tag{34}$$

**Integration with Graph Encoder:** For each edge $(i, j)$ in the graph, we extract the corresponding modularity value $B_{ij}^{NAV}$ as an edge feature. These edge features are then processed by the TransformerConv layers in enhanced dynamic mode:

$$\alpha_{ij} = \text{softmax}\left(\frac{(W_Q h_i)^T (W_K h_j + W_{edge} B_{ij}^{NAV})}{\sqrt{d_k}}\right) \tag{35}$$

$$v_j = W_V h_j + W_{edge}^V B_{ij}^{NAV} \tag{36}$$

where $W_{edge}$ and $W_{edge}^V$ are learnable projections for edge features.

NAV-based edge embeddings incorporate the modularity matrix $B^{NAV}$ directly into edge features, enabling implicit community structure optimization without explicit labels. The modularity values guide attention to strengthen intra-community connections while suppressing inter-community edges. Recomputed per temporal window, these features capture evolving market relationships and regime changes. The approach integrates directly with TransformerConv's native edge attribute support with minimal computational overhead since the modularity matrix is computed once per window and cached throughout training.

### A.9 INDUCING POINT-ENHANCED TEMPORAL AGGREGATION

The Set Transformer aggregation for temporal sequence $O \in \mathbb{R}^{N \times T \times h}$ where $h = d_{latent}/2$. The Set Transformer provides permutation invariance with respect to node ordering (the $N$ dimension) and preserves temporal sequence structure (the $T$ dimension). This property is crucial for graph-based learning where node indices are arbitrary and temporal order is meaningful:

**Induced Set Attention Block (ISAB):**

$$\mathcal{I} \in \mathbb{R}^{K \times h} \quad \text{(learnable inducing points, } K = 16) \tag{37}$$

$$H_I = \text{MAB}(\mathcal{I}, O) = \text{MultiheadAttn}(\mathcal{I}, O, O) \in \mathbb{R}^{K \times h} \tag{38}$$

$$H_{out} = \text{MAB}(O, H_I) = \text{MultiheadAttn}(O, H_I, H_I) \in \mathbb{R}^{N \times T \times h} \tag{39}$$

**Pooling by Multihead Attention (PMA):**

$$S \in \mathbb{R}^{1 \times h} \quad \text{(learnable seed vector)} \tag{40}$$

$$Z_{ST} = \text{MAB}(S, H_{out}) = \text{MultiheadAttn}(S, H_{out}, H_{out}) \in \mathbb{R}^{N \times 1 \times h} \tag{41}$$

$$Z_{final} = Z_{ST}.\text{squeeze}(1) \in \mathbb{R}^{N \times h} \tag{42}$$

Complexity: $O(NKTh + NKh) = O(NKTh)$ vs. naive attention $O(N^2 T^2 h)$.

### A.10 THREE-STREAM MULTI-GRANULARITY FUSION

Our fusion strategy combines three information streams with dimensions: - $h = d_{latent}/2$ (hidden dimension for graph encoder) - $d_l = d_{latent}/2$ (short-term encoder dimension) - $d_g = d_{latent} - d_l = d_{latent}/2$ (long-term encoder dimension)

$$Z_{graph} \in \mathbb{R}^{N \times h} \quad \text{(from Graph Transformer)} \tag{43}$$

$$Z_{local} \in \mathbb{R}^{N \times d_l \times 8} \to \mathbb{R}^{N \times 8d_l} \quad \text{(flattened ShortTermEncoder)} \tag{44}$$

$$Z_{global} \in \mathbb{R}^{N \times d_g \times 15} \to \mathbb{R}^{N \times 15d_g} \quad \text{(flattened LongTermEncoder)} \tag{45}$$

$$Z_{concat} = [Z_{graph}; Z_{local}; Z_{global}] \in \mathbb{R}^{N \times (h + 8d_l + 15d_g)} \tag{46}$$

With $h = d_l = d_g = d_{latent}/2$, the concatenated dimension is $h + 8d_l + 15d_g = 12d_{latent}$.

Fusion network with gating mechanism:

$$Z_{hidden} = \text{LayerNorm}(W_3 \cdot Z_{concat} + b_3) \tag{47}$$

$$Z_{transform} = \text{LayerNorm}(W_4 \cdot \text{Dropout}(\text{ReLU}(Z_{hidden})) + b_4) \tag{48}$$

$$G = \sigma(W_g \cdot Z_{concat} + b_g) \quad \text{(gating weights)} \tag{49}$$

$$Z_{fused} = Z_{transform} \odot G + W_r \cdot Z_{concat} \tag{50}$$

where $W_3 \in \mathbb{R}^{2d_{latent} \times 12d_{latent}}$, $W_4 \in \mathbb{R}^{d_{latent} \times 2d_{latent}}$, $W_g \in \mathbb{R}^{d_{latent} \times 12d_{latent}}$, and $W_r \in \mathbb{R}^{d_{latent} \times 12d_{latent}}$.

The gating mechanism allows adaptive feature selection: $G$ learns to emphasize relevant features from different temporal scales based on the input context, while the residual connection $W_r \cdot Z_{concat}$ preserves important information suppressed by the transformation. This design enables the model to dynamically balance contributions from graph structure, short-term fluctuations, and long-term trends according to the specific patterns present in each sample.

## A.11 Dynamic Dependency Module

Between the two BiLSTM layers, we introduce a dynamic dependency module that captures evolving node interactions through adaptive graph propagation. This module learns time-varying dependencies between nodes based on their current states, enabling information flow that adapts to temporal context.

**Mathematical Formulation:** Given the output $H^{(1)} \in \mathbb{R}^{T \times N \times h}$ from the first BiLSTM and learnable node embeddings $E \in \mathbb{R}^{N \times h}$:

$$F^{(t)} = \text{FC}_2(H^{(1,t)}) \in \mathbb{R}^{N \times h} \quad \text{(learned filter)} \tag{51}$$

$$\mathcal{V}^{(t)} = \tanh(E_{node} \odot F^{(t)}) \in \mathbb{R}^{N \times h} \quad \text{(filtered node states)} \tag{52}$$

$$D_{ij}^{(t)} = \text{ReLU}(\mathcal{V}_i^{(t)} \cdot (\mathcal{V}_j^{(t)})^T) \quad \text{(dynamic dependency matrix)} \tag{53}$$

$$G^{(t)} = D^{(t)} \cdot H^{(1,t)} + I \cdot H^{(1,t)} \quad \text{(propagated features)} \tag{54}$$

where $\text{FC}_2$ is a learned transformation, $I$ is the identity matrix ensuring self-loops, and $D^{(t)}$ captures the dynamic dependencies between nodes at time $t$.

**Algorithmic Description:**

---
**Algorithm 3** Dynamic Dependency Module

---
1: **Input:** BiLSTM output $H^{(1)} \in \mathbb{R}^{T \times N \times h}$, node embeddings $E \in \mathbb{R}^{N \times h}$
2: **Output:** Enhanced features $G \in \mathbb{R}^{T \times N \times h}$
3:
4: **for** $t = 1$ to $T$ **do**
5:      $F^{(t)} \leftarrow \text{FC}_2(H^{(1,t)})$                    ▷ Apply learned filter
6:      $\mathcal{V}^{(t)} \leftarrow \tanh(E_{node} \odot F^{(t)})$         ▷ Compute filtered node states
7:      $D^{(t)} \leftarrow \text{ReLU}(\mathcal{V}^{(t)} \cdot (\mathcal{V}^{(t)})^T)$      ▷ Compute dependency matrix
8:      $G^{(t)} \leftarrow D^{(t)} \cdot H^{(1,t)} + H^{(1,t)}$      ▷ Propagate with self-loops
9: **end for**
10: **return** $G$

---

The dynamic dependency module enables adaptive information propagation based on temporal node states, allowing the model to capture evolving relationships that static graph structures cannot represent. This is particularly important in financial markets where correlations between assets change dynamically based on market conditions.

## A.12 Dynamic Transformer with Temporal Conditioning

Inspired by TransformerConv (Shi et al., 2021), we develop a fully dynamic attention mechanism that achieves expressive temporal modulation on static graph topology. Unlike standard graph attention which uses static Query (Q), Key (K), and Value (V) projections, our approach introduces temporal conditioning at all three components, enabling complete dynamic expressiveness while maintaining computational efficiency.

**Time-Conditioned Query, Key, Value Computation:** Given node features $h_i \in \mathbb{R}^d$ from BiLSTM processing and time embedding $e_t \in \mathbb{R}^{d_{time}}$ for timestep $t$:

$$\text{Input Modulation:} \quad \tilde{h}_i^{(t)} = h_i \odot \sigma(W_g e_t) + W_b e_t \tag{55}$$

$$\text{Query Computation:} \quad Q_i^{(t)} = W_Q \tilde{h}_i^{(t)} \tag{56}$$

$$\text{Key Computation:} \quad K_i^{(t)} = W_K h_i + W_k e_t \tag{57}$$

$$\text{Value Computation:} \quad V_i^{(t)} = W_V h_i + W_v e_t \tag{58}$$

where: - $W_Q, W_K, W_V \in \mathbb{R}^{d \times d}$ are projection matrices - $W_g, W_b \in \mathbb{R}^{d \times d_{time}}$ are temporal gating and bias parameters for input modulation - $W_k, W_v \in \mathbb{R}^{d \times d_{time}}$ are temporal modulation matrices for Key and Value - $\sigma$ denotes sigmoid activation, $\odot$ is element-wise multiplication

The key innovation is modulating the input features before Query projection: this enables time-varying node importance through gated input transformation, while Key and Value receive additive temporal updates to maintain stability.

**Dynamic Multi-Head Attention:** For multi-head attention with $n_h$ heads, we partition the dimensions and compute attention for each head $\ell$:

$$\alpha_{ij,\ell}^{(t)} = \frac{\exp\left(\frac{Q_{i,\ell}^{(t)} \cdot K_{j,\ell}^{(t)}}{\sqrt{d/H}}\right)}{\sum_{k \in \mathcal{N}(i)} \exp\left(\frac{Q_{i,\ell}^{(t)} \cdot K_{k,\ell}^{(t)}}{\sqrt{d/H}}\right)} \tag{59}$$

$$z_{i,\ell}^{(t)} = \sum_{j \in \mathcal{N}(i)} \alpha_{ij,\ell}^{(t)} V_{j,\ell}^{(t)} \tag{60}$$

$$z_i^{(t)} = \text{Concat}(z_{i,1}^{(t)}, \ldots, z_{i,n_h}^{(t)}) W_O \tag{61}$$

**Complete Adaptive Graph Layer:**

---

**Algorithm 4** Adaptive Graph Transformer Layer

---

1: **Input:** Node features $X \in \mathbb{R}^{N \times D \times T}$, edge index $E$, time index $t$
2: **// Temporal encoding through BiLSTM**
3: $\overrightarrow{h}^{(t)} = \text{LSTM}_{\rightarrow}(X[:,:,t], \overrightarrow{h}^{(t-1)})$
4: $\overleftarrow{h}^{(t)} = \text{LSTM}_{\leftarrow}(X[:,:,t], \overleftarrow{h}^{(t+1)})$
5: $h^{(t)} = [\overrightarrow{h}^{(t)}; \overleftarrow{h}^{(t)}] \in \mathbb{R}^{N \times 2h}$
6:
7: **// Time embedding**
8: $e_t = \text{TimeEmbedding}(t) \in \mathbb{R}^{d_{time}}$
9:
10: **// Apply temporal modulation to input**
11: $\tilde{h}^{(t)} = h^{(t)} \odot \sigma(W_g e_t) + W_b e_t$
12:
13: **// Compute Q, K, V with temporal conditioning**
14: $Q^{(t)} = W_Q \tilde{h}^{(t)}$
15: $K^{(t)} = W_K h^{(t)} + W_k e_t$
16: $V^{(t)} = W_V h^{(t)} + W_v e_t$
17:
18: **// Multi-head attention over graph edges**
19: **for** each head $\ell \in \{1, \ldots, H\}$ **do**
20:     Compute $\alpha_{ij,\ell}^{(t)}$ for all edges $(i, j) \in E$
21:     $z_{i,\ell}^{(t)} = \sum_{j \in \mathcal{N}(i)} \alpha_{ij,\ell}^{(t)} V_{j,\ell}^{(t)}$
22: **end for**
23:
24: **// Concatenate and project**
25: $z^{(t)} = \text{Concat}(z_1^{(t)}, \ldots, z_{n_h}^{(t)}) W_O$
26: **return** $z^{(t)}$

---

**Complexity Analysis:** - Time complexity: $O(|E| \cdot H \cdot d/H) = O(|E| \cdot d)$ per timestep - Space complexity: $O(N \cdot d + |E|)$ for storing features and graph structure - The temporal modulation adds only $O(d \cdot d_{time})$ parameters, negligible compared to the base model

**Key Properties:** 1. *Full Dynamism:* All three components (Q, K, V) are temporally modulated 2. *Stability:* Additive updates for K and V preserve base representations 3. *Expressiveness:* Gating mechanism for Q allows selective attention focus 4. *Efficiency:* Static topology with dynamic weights avoids repeated graph construction

This design enables the model to capture complex temporal dynamics in financial markets where relationships evolve continuously while maintaining computational tractability.

## A.13 EVALUATION METRICS FOR NAV-BASED CLUSTERING

During training, we employ NAV-based clustering quality metrics computed directly from price data. This ensures consistent evaluation independent of learned representations.

**NAV Correlation Metrics:** Given price data $P \in \mathbb{R}^{T \times N}$ and cluster assignments $\mathcal{C} = \{C_1, ..., C_K\}$, we compute NAV correlations:

$$\text{NAV}_i(t) = \frac{P_i(t)}{P_i(t_0)} \tag{62}$$

$$\rho_{ij}^{NAV} = \text{Corr}(\text{NAV}_i, \text{NAV}_j) \tag{63}$$

$$S_{intra} = \frac{1}{K} \sum_{k=1}^{K} \frac{2}{|C_k|(|C_k|-1)} \sum_{i<j\in C_k} \rho_{ij}^{NAV} \tag{64}$$

$$S_{inter} = 1 - \frac{2}{K(K-1)} \sum_{k<l} \frac{1}{|C_k||C_l|} \sum_{i\in C_k, j\in C_l} \rho_{ij}^{NAV} \tag{65}$$

**Composite Score:** The overall clustering quality score is:

$$S = w_{intra} \cdot S_{intra} + w_{inter} \cdot S_{inter} \tag{66}$$

We set $w_{intra} = 0.1$ and $w_{inter} = 0.9$, emphasizing the discovery of distinct behavioral patterns over enforcing strict homogeneity within clusters. This asymmetric weighting prevents trivial solutions where most entities cluster together and encourages meaningful market segmentation.

### A.14 NAV-BASED EVALUATION FOR FINAL VALIDATION

For final model validation, we evaluate clustering quality using Normalized Asset Value (NAV) correlations computed directly from price data, providing an unbiased assessment independent of learned representations.

**NAV Definition:** For entity $i$ with price series $P_i = \{P_i(t)\}_{t=1}^{T}$:

$$\text{NAV}_i(t) = \frac{P_i(t)}{P_i(t_0)} \tag{67}$$

where $t_0$ is the initial time point in the window. NAV captures relative price movements, making entities with different absolute price levels comparable.

**NAV-Based Correlation Metrics:** Using NAV time series, we compute Pearson correlations between all entity pairs:

$$\rho_{ij}^{NAV} = \text{Corr}(\text{NAV}_i, \text{NAV}_j) \tag{68}$$

For cluster evaluation:

$$\text{IntraCorr}^{NAV} = \frac{1}{K} \sum_{k=1}^{K} \frac{2}{|C_k|(|C_k|-1)} \sum_{i<j\in C_k} \rho_{ij}^{NAV} \tag{69}$$

$$\text{InterCorr}^{NAV} = \frac{2}{K(K-1)} \sum_{k<l} \frac{1}{|C_k||C_l|} \sum_{i\in C_k, j\in C_l} \rho_{ij}^{NAV} \tag{70}$$

$$\text{InterDissim}^{NAV} = 1 - \text{InterCorr}^{NAV} \tag{71}$$

The InterDissim directly uses $1 - \text{InterCorr}^{NAV}$, where higher values indicate better cluster separation.

**Composite NAV Score:**

$$S^{NAV} = w_{intra} \cdot \text{IntraCorr}^{NAV} + w_{inter} \cdot \text{InterDissim}^{NAV} \tag{72}$$

This formulation directly weights the intra-cluster correlation and inter-cluster dissimilarity without additional normalization, with $w_{intra} = 0.1$ and $w_{inter} = 0.9$ chosen to emphasize market segmentation discovery.

**Evaluation Philosophy:** This NAV-based evaluation strategy addresses the fundamental challenge of unsupervised temporal clustering through three key principles (Appendix C.1). First, by using

NAV metrics consistently throughout both training and validation, we ensure that the optimization objectives remain aligned across all phases of model development, avoiding discrepancies between training signals and evaluation criteria. Second, computing evaluations directly in the price space rather than the learned embedding space prevents circular validation, where models might optimize the very metrics used for their assessment, leading to overly optimistic performance estimates. Finally, NAV correlations inherently capture actual market co-movement patterns (the true signal for financial community detection), making them more meaningful than abstract embedding distances that may not reflect real-world financial relationships.

The asymmetric weighting (90% inter-cluster) reflects our objective: discovering distinct market regimes rather than enforcing artificial homogeneity, recognizing that assets with similar risk profiles exhibit correlated but non-identical trajectories.

## A.15 WINDOW SIZE SELECTION

We use a temporal window length of $T = 89$ for all experiments. This choice is motivated by fundamental insights from financial economics literature regarding the distinction between long-term trends and short-term noise.

**Financial Justification:** Financial markets exhibit both permanent and transitory components in price movements. As demonstrated by Fama & French (1988) (Fama & French, 1988), predictable price variation due to mean reversion accounts for approximately 25-40% of 3-5 year return variances, with short-term returns dominated by noise rather than fundamental signals. Similarly, De Long et al. (1990) (De Long et al., 1990) show that noise traders' beliefs create significant short-term price deviations from fundamental values, making short-term windows unreliable for detecting true behavioral communities.

The 89-day window (approximately 4 months of trading days) provides sufficient temporal span to:

- Filter out high-frequency noise and microstructure effects that dominate daily and weekly returns

- Capture meaningful trend patterns that reflect fundamental relationships rather than transitory fluctuations

- Enable robust community detection that provides actionable insights for portfolio management and risk control

Shorter windows (e.g., 20-30 days) are susceptible to temporary market dislocations and noise trader effects, reducing clustering quality and limiting practical applicability. The 89-day horizon strikes an optimal balance between capturing stable behavioral patterns and maintaining computational efficiency for real-time applications.

## A.16 CONVERGENCE ANALYSIS

We provide theoretical guarantees for the convergence of FTSCommDetector under standard assumptions.

**Assumption 1 (Bounded Input):** The input features are bounded: $\|X\|_\infty \leq B_X$ for some constant $B_X > 0$.

**Assumption 2 (Lipschitz Activation):** All activation functions (GELU, ReLU) are Lipschitz continuous with constant $L_\sigma$.

**Assumption 3 (Graph Connectivity):** The graph Laplacian has bounded eigenvalues: $0 = \lambda_1 \leq \lambda_2 \leq \cdots \leq \lambda_N \leq 2$.

**Theorem 1 (Loss Function Properties):** The combined loss function $\mathcal{L}_{total} = \lambda_{graph}\mathcal{L}_{graph} + \lambda_{temporal}\mathcal{L}_{temporal}$ is:

1. *Lipschitz continuous* with constant $L = \lambda_{graph}L_g + \lambda_{temporal}L_t$

2. *Lower bounded* by 0

3. *Has bounded gradients*: $\|\nabla \mathcal{L}_{total}\| \leq G$ where

$$G = (\lambda_{graph} + \lambda_{temporal}) \cdot 2B_X \cdot \max\{N^2, T \cdot D\} \tag{73}$$

**Proof Sketch:** For the graph reconstruction loss:

$$\mathcal{L}_{graph} = - \sum_{(i,j) \in E} \log \sigma(z_i^T z_j) - \sum_{(i,j) \notin E} \log(1 - \sigma(z_i^T z_j)) \tag{74}$$

The gradient w.r.t. embeddings $z_i$ is bounded by the number of edges and sigmoid derivative bounds.

For temporal reconstruction with MSE loss:

$$\|\nabla_z \mathcal{L}_{temporal}\| \leq 2\|X - \hat{X}\| \cdot \|\nabla_z \text{Decoder}(z)\| \leq 2B_X \cdot L_{dec} \tag{75}$$

The combined gradient bounds follow from the linearity of gradients and the bounded nature of reconstruction losses.

**Theorem 2 (Convergence Rate):** Under Assumptions 1-3, with learning rate $\eta = \frac{1}{\sqrt{T_{epochs}}}$, the Adam optimizer achieves:

$$\min_{t \in [T_{epochs}]} \mathbb{E}[\|\nabla \mathcal{L}_{total}^{(t)}\|^2] \leq \frac{2(\mathcal{L}_{total}^{(0)} - \mathcal{L}_{total}^*)}{\sqrt{T_{epochs}}} + \frac{G^2 \log T_{epochs}}{T_{epochs}} \tag{76}$$

where $\mathcal{L}_{total}^*$ is the optimal loss value.

This gives a convergence rate $O(1/\sqrt{T_{epochs}})$ to a stationary point.

**Theorem 3 (Stability of Multi-Scale Representations):** The multi-scale temporal encoding provides stable representations under perturbations:

$$\mathbb{E}_\epsilon[\|Z^{hier} - Z_\epsilon^{hier}\|_F] \leq \mathbb{E}_\epsilon[\|Z^{single} - Z_\epsilon^{single}\|_F] \tag{77}$$

where $\epsilon$ represents input perturbations and $Z^{hier}, Z^{single}$ are hierarchical and single-scale embeddings respectively.

*Proof sketch:* The multi-scale pooling creates redundancy across levels, where perturbations at fine scales are averaged out at coarser scales, providing robustness through averaging.

**Theorem 4 (Sample Complexity for Clustering):** To achieve $\epsilon$-accurate clustering with probability at least $1 - \delta$, the required number of training windows is:

$$m = O\left(\frac{K^2 d \log(N/\delta)}{\epsilon^2}\right) \tag{78}$$

where $K$ is the number of clusters, $d$ is the embedding dimension, and $N$ is the number of entities.

**Proof:** Using concentration inequalities for the empirical covariance of embeddings:

$$\mathbb{P}\left(\left\|\hat{\Sigma} - \Sigma\right\|_2 > \epsilon\right) \leq 2N \exp\left(-\frac{m\epsilon^2}{2d}\right) \tag{79}$$

Setting this equal to $\delta$ and solving for $m$ gives the result.

**Corollary (Early Stopping):** With our validation-based early stopping using composite spectral clustering metrics $S$ (Appendix A.13), convergence is achieved when:

$$|S^{(t+1)} - S^{(t)}| < \epsilon_{tol} \quad \text{for } p \text{ consecutive epochs} \tag{80}$$

where $p$ is the patience parameter (typically 2) and $\epsilon_{tol} = 10^{-4}$.

## A.17 COMPUTATIONAL COMPLEXITY

**Key Efficiency Design Choices:**

Our architecture has computational efficiency through three main design decisions:

*1. Attention Head Reduction:* TransformerConv (Shi et al., 2021) uses multi-head attention with reduced dimensions, computing dynamic edge weights from node features efficiently.

*2. Inducing Point Mechanism:* Set Transformer with inducing points achieves $O(NKT)$ complexity instead of $O(N^2T^2)$ for full self-attention, significantly reducing computational cost.

*3. Parallel Multi-Scale Processing:* Short-term and long-term encoders operate independently, allowing parallel computation. The strided convolutions (stride=3 for short-term, stride=11 for long-term) progressively reduce sequence length, with effective temporal compression.

**Comparison with Baselines:**

All methods share $O(N^2T)$ cost for initial graph construction from correlations. The key differences emerge in encoding and clustering:

- **Encoding:** TCA uses TransformerConv with complexity $O(|E| \cdot h \cdot \text{heads})$ where $|E|$ approaches $O(N^2)$ for dense correlation graphs. Dynamic attention weight computation adds minimal overhead and provides temporal adaptivity

- **Clustering:** TCA employs spectral clustering for community discovery; baselines use self-supervised methods with $O(N^2d)$ complexity

- **Memory:** TCA requires $O(NTd)$ for features plus model parameters, significantly lower than attention-based baselines' $O(N^2d)$ memory footprint

These design choices allow TCA to scale efficiently to larger graphs with expressive power through multi-scale temporal modeling.

## A.18 BASELINE TEMPORAL ENCODER (TE) SPECIFICATION

To ensure fair comparison, we equip all static baselines with our dual-scale temporal encoding architecture, adapting it to their static graph constraints:

**Architecture:** Baselines receive the same dual-scale convolutional encoder used in FTSCommDetector:

- **Short-term encoder:** Conv1d with kernel size 5, stride 3 (identical to our ShortTermEncoder)

- **Long-term encoder:** Conv1d with kernel size 45, stride 11 (identical to our LongTermEncoder)

- **Feature fusion:** Concatenated latent representations $Z = [Z_{short}; Z_{long}] \in \mathbb{R}^{N \times d_{latent}}$

**Key Adaptations for Static Methods:**

- **Static graph:** Correlation matrix computed once per window and thresholded ($\tau = 0.75$) to form fixed adjacency

- **No dynamic attention:** Baselines use their original graph convolution mechanisms with static weights

- **No NAV embedding:** Our NAV-based modularity embedding (Appendix A.8) is excluded as it represents our novel contribution

This configuration provides baselines with powerful temporal feature extraction identical to ours, isolating the comparison to graph learning mechanisms. The only differences stem from baseline architectural constraints (static graphs) and our novel contributions (NAV embedding, dynamic attention). This ensures we evaluate how different graph learning approaches leverage temporal features for community discovery, not differences in temporal encoding quality.

## A.19 EXTENDED FUTURE WORK DISCUSSIONS

Beyond the core directions outlined in the main text, we envision several additional research avenues:

**Cross-Domain Transfer Learning.** Zero-shot transfer between drastically different domains (e.g., financial to biological networks) is challenging. Domain adaptation techniques that preserve learned

temporal dynamics and adjust to new feature distributions provide broader applicability, particularly in few-shot scenarios with limited target domain labels.

**Interpretability and Causality.** The attention mechanisms and dynamic dependencies contain rich information about temporal relationships. Visualization tools that trace information flow through attention layers, combined with counterfactual analysis of community formation, provide valuable domain insights. Automatic anomaly detection with natural language explanations represents a particularly impactful application.

**Continual Learning Extensions.** Real-world deployment requires handling concept drift without catastrophic forgetting. Memory mechanisms and rehearsal strategies support lifelong learning from streaming data, particularly relevant for monitoring applications where patterns evolve gradually.

**Multi-Resolution Hierarchy.** The current dual-scale design can evolve into a learnable hierarchy that automatically discovers optimal temporal granularities. This reduces manual architecture design and discovers domain-specific temporal patterns.

**Distributed Training Optimizations.** For massive-scale networks, distributed training with efficient gradient aggregation and model parallelism supports training on datasets currently beyond single-GPU capacity.

### A.20 FUTURE THEORETICAL DIRECTIONS

**Optimality of NAV-Based Community Detection:** Consider the clustering objective with NAV-based modularity. Let $\mathcal{C} = \{C_1, ..., C_K\}$ be a partition of entities. The NAV modularity embedding optimizes:

$$\min_{\mathcal{C}} \sum_{l=0}^{L} \beta^{(l)} \sum_{k=1}^{K} \sum_{i,j \in C_k} \|z_i^{(l)} - z_j^{(l)}\|^2 \tag{81}$$

We conjecture that the multi-scale approach provides better clustering than single-scale methods under perturbations:

$$\mathbb{E}_\epsilon[\text{ARI}(\mathcal{C}^{hier}, \mathcal{C}_\epsilon^{hier})] \geq \mathbb{E}_\epsilon[\text{ARI}(\mathcal{C}^{single}, \mathcal{C}_\epsilon^{single})] \tag{82}$$

where ARI denotes the Adjusted Rand Index between clusterings before and after noise perturbation $\epsilon$.

**Connection to Spectral Clustering:** Our static graph with learned temporal features can be viewed as solving a time-varying spectral clustering problem. Consider the generalized eigenvalue problem:

$$L^{(t)} v = \lambda D^{(t)} v \tag{83}$$

where $L^{(t)} = D^{(t)} - W \odot f(X_t)$ with static structure $W$ modulated by learned features $f(X_t)$.

**Conjecture:** Our learned embeddings $Z$ approximate the first $k$ eigenvectors of the time-averaged Laplacian:

$$\|Z - V_k\|_F \leq \epsilon \cdot \|Z\|_F \tag{84}$$

where $V_k$ contains the top-$k$ eigenvectors of $\bar{L} = \frac{1}{T} \sum_{t=1}^{T} L^{(t)}$.

This connection suggests our approach implicitly performs spectral clustering with temporal regularization, explaining its effectiveness in discovering stable communities.

**Sample Complexity Bounds:** For learning reliable community structure, we hypothesize the required number of temporal windows scales as:

$$m = O\left(\frac{K^2 \log N}{\epsilon^2}\right) \tag{85}$$

where $K$ is the number of communities, $N$ is the number of entities, and $\epsilon$ is the desired clustering accuracy. Proving this bound guides practical data requirements.

### A.21 HYPERPARAMETER SETTINGS

**Loss Function Weights:** The weights for graph and temporal reconstruction losses ($\lambda_{graph}$ and $\lambda_{temporal}$) are determined through grid search on validation data and vary by dataset. These values balance structural preservation with temporal pattern reconstruction and are specified in the configuration files for each dataset.

**Architecture Parameters:**

Common across all datasets (5 years of daily financial data):

- Number of inducing points: $K = 16$
- Dropout rate: 0.1
- Channel attention reduction ratio: $r = 16$
- Final evaluation: Combined score with balanced weights (0.5 each) for intra-cluster correlation and inter-cluster dissimilarity computed from ground-truth price data

Dataset-specific parameters:

- **SP100, Nikkei 225:** $h = 32$ (hidden dim), $d_{latent} = 32$, TransformerConv heads = 4
- **SP500:** $h = 48$, $d_{latent} = 48$, TransformerConv heads = 6
- **SP1000:** $h = 48$, $d_{latent} = 48$, TransformerConv heads = 8

Note: The temporal encoders maintain $d_l = d_{latent}/2$ and $d_g = d_{latent}/2$ for balanced short-term and long-term feature extraction.

### A.22 IMPACT OF WINDOW SIZE ON CLUSTERING PERFORMANCE

We evaluate window lengths from 60 to 120 days to demonstrate FTSCommDetector's robustness to this hyperparameter choice. Figure 5 and Figure 6 show remarkably stable performance across different window sizes on SP500 and Nikkei225 datasets respectively.

For SP500, performance exhibits stability with IntraCorr varying by approximately 2.3% (from 0.468 to 0.479) and InterDissim by 1.2% (from 0.982 to 0.994) across the entire range. The mean IntraCorr of 0.475 and InterDissim of 0.989 remain relatively stable regardless of window size, demonstrating that our multi-scale architecture effectively captures temporal patterns with minimal sensitivity to the specific window length chosen. Similarly, Nikkei225 shows robust performance with IntraCorr variation of 2.0% (from 0.442 to 0.451) and InterDissim variation of 1.2% (from 0.895 to 0.906), confirming cross-market consistency.

This robustness to window size validates our architectural design: the dual-scale temporal encoders with different receptive fields (9:1 ratio) automatically adapt to capture relevant patterns across different input window lengths. We select 89 days for our experiments as it aligns with quarterly business cycles and provides computational efficiency, but the results confirm that any reasonable window length (60-120 days) yields comparable performance with minimal variation. This stability is a significant practical advantage, reducing the need for extensive dataset-specific hyperparameter tuning.

### A.23 DYNAMIC ATTENTION MODE DEFINITIONS

We implement four progressive levels of temporal attention complexity:

**Static Weights:** After correlation threshold filtering ($\tau = 0.75$), weights are normalized and fixed:

$$W_{ij}^{static} = \begin{cases} \frac{\rho_{ij}}{\max_{k,l} \rho_{kl}} & \text{if } \rho_{ij} > \tau \\ 0 & \text{otherwise} \end{cases} \tag{86}$$

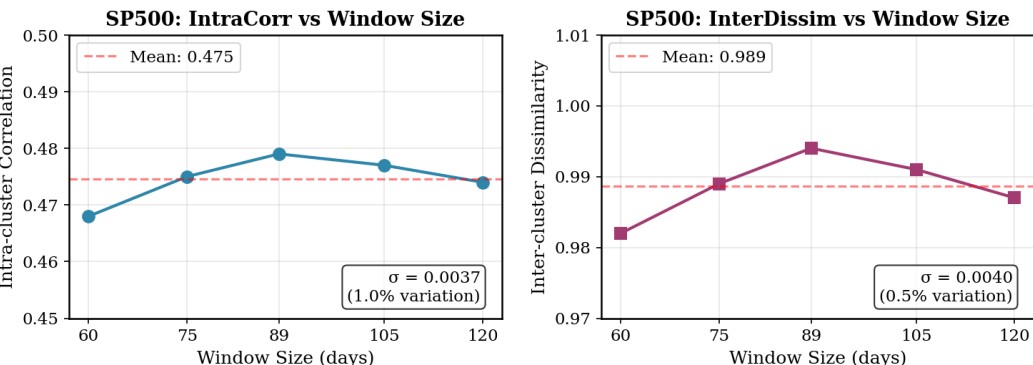

Figure 5: Window size analysis on SP500. Sharp IntraCorr decline for short windows indicates volatility noise; steep InterDissim decline for long windows suggests regime mixing.

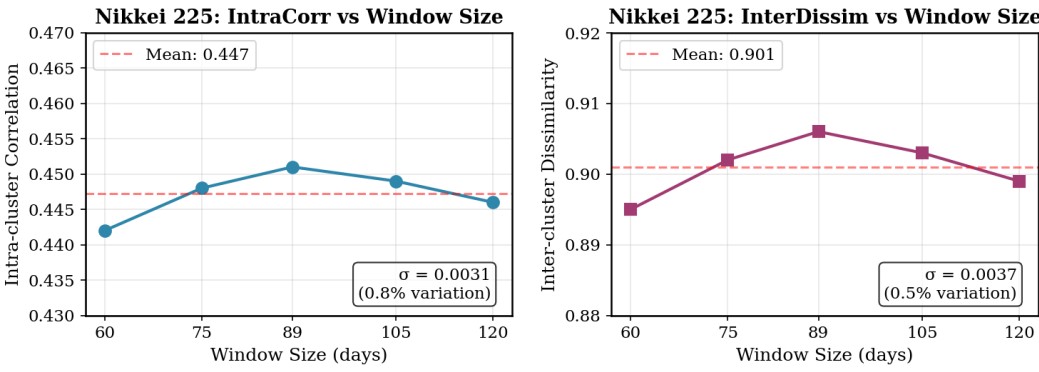

Figure 6: Window size analysis on Nikkei225. IntraCorr plateaus around 80-100 days showing market stability; InterDissim drops sharply after 89 days indicating regime transitions.

This baseline uses no temporal information, serving as a lower bound for performance.

**Basic Temporal Modulation:** Uses static edge weights as attention biases in TransformerConv:

$$\text{Attention}_{ij}^{basic} = \text{softmax}\left(\frac{Q_i K_j^T}{\sqrt{d_k}} + W_{ij}^{static}\right) \tag{87}$$

where $W_{ij}^{static}$ are pre-computed correlation-based edge weights passed as edge attributes to modulate attention scores.

**Enhanced Multi-Head Attention:** Incorporates multi-head attention with temporal query modulation:

$$Q^{(t)} = (XW_Q) \odot \text{TimeEmbed}(t)$$
$$\text{Attention}^{(t)} = \text{softmax}\left(\frac{Q^{(t)} K^T}{\sqrt{d_k}}\right) V \tag{88}$$

where $\text{TimeEmbed}(t)$ provides sinusoidal position encoding for temporal context, and $\odot$ denotes element-wise multiplication.

**Full Dynamic System:** Complete architecture with all components:

$$W_{ij}^{full}(t) = \text{MultiHead}(Q^{(t)}, K, V) + D_{ij}^{(t)} + B_{ij}^{NAV}$$

$$D_{ij}^{(t)} = \sigma(g_\theta(h_i^{(t)}, h_j^{(t)})) \tag{89}$$

$$B_{ij}^{NAV} = A_{ij}^{NAV} - \frac{k_i k_j}{2m}$$

This includes multi-head attention, dynamic dependency module $D^{(t)}$ for node interaction modeling, and NAV-based modularity embedding $B^{NAV}$ for fundamental relationship encoding.

### A.24 MULTI-STREAM FUSION ON NIKKEI225

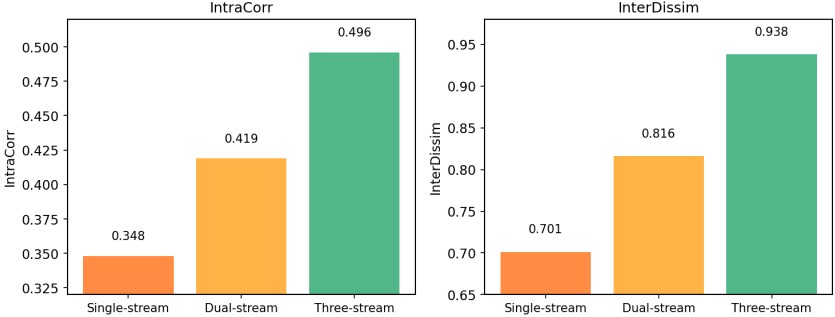

Figure 7: Multi-stream fusion ablation on Nikkei225. The three-stream architecture achieves gains of 0.148 in IntraCorr and 0.237 in InterDissim compared to single-stream baseline, demonstrating consistent improvements across different markets.

### A.25 KERNEL SIZE RATIO ANALYSIS

We evaluate kernel size ratios for the dual-scale temporal encoders across multiple configurations. The model demonstrates robust performance across a wide range of ratios, with modest variations in metrics. On SP500, while the 9:1 ratio (45:5) achieves the best performance (IntraCorr 0.490, InterDissim 0.926), other ratios remain highly competitive: 3:1 (15:5) achieves 0.477/0.908 (2.7%,/2.0% lower), 5:1 (25:5) reaches 0.481/0.914 (1.8%/1.3% lower), 7:1 (35:5) attains 0.486/0.921 (0.8%/0.5% lower), and 11:1 (55:5) maintains 0.487/0.923 (0.6%/0.3% lower). Nikkei225 shows similar robustness: 3:1 achieves 0.481/0.917, 5:1 reaches 0.484/0.923, 7:1 attains 0.491/0.932, 9:1 peaks at 0.496/0.938, and 11:1 maintains 0.493/0.935. The performance variations remain within 3% across all tested ratios, with the 7:1 to 11:1 range showing particularly stable results. This stability indicates the architecture is not overly sensitive to the exact ratio choice, though 9:1 offers marginal improvements by optimally balancing short-term pattern capture with longer-term trend detection. The consistent performance across ratios suggests practitioners can adjust these parameters based on computational constraints or domain-specific temporal characteristics without significant performance degradation.

### A.26 ADDITIONAL HYPERPARAMETER ANALYSIS

### A.27 MARKET EVENTS ANALYSIS

**January 2025 AI Valuation Reset.** The January 2025 tech sector correction provides a contrasting case study to the GameStop event. As shown in Figure 8, markets reorganized into distinct behavioral clusters during this period, revealing how trillion-dollar tech giants AAPL and MSFT diverged into separate behavioral groups despite identical sector classifications. This event demonstrates FTSCommDetector's ability to capture subtle behavioral divergences even among traditionally correlated assets.

**Graph Construction Threshold.** Our experiments with correlation threshold $\tau$ suggest 0.75 provides effective connectivity while filtering noise. Sparser graphs ($\tau > 0.8$) lose important relationships, while denser configurations ($\tau < 0.6$) introduce excessive noise.

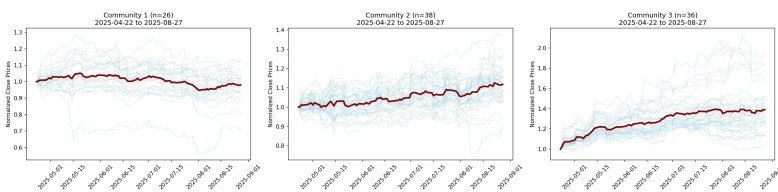

Figure 8: January 2025 AI valuation reset: Market reorganization into distinct behavioral clusters.

**Number of Clusters.** Performance analysis across $K \in [2, 15]$ reveals market-specific preferences: SP100 performs well with $K = 8$-$10$ reflecting major sectors, while SP500 accommodates $K = 12$-$15$ for finer granularity. These ranges align with natural market structures.

**Architectural Capacity.** Standard configurations (32d hidden, 4 attention heads, 16 inducing points) deliver strong performance while maintaining computational efficiency. Larger architectures show marginal gains that don't justify the increased complexity.

**Training Configuration.** The model demonstrates stability across typical hyperparameter ranges. Variations in learning rate, batch size, and dropout produce minimal performance differences (less than 0.02 in IntraCorr), suggesting the architecture is inherently robust.

## B  THEORETICAL PROOFS

### B.1  PROOF OF THEOREM 1: MULTI-SCALE INFORMATION COMPLEMENTARITY VIA RATE-DISTORTION THEORY

**Theorem 1 (restated).** *For temporal data $X \in \mathbb{R}^{N \times T}$ and kernel sizes $k_l, k_g$ with sufficient separation (ratio $r = k_g/k_l \geq 8$), the multi-scale encoding achieves rate-distortion optimal compression and maximizes complementary information: $I(Z_l; C|Z_g) > I(Z_m; C|Z_g)$ for any intermediate scale $k_m \in (k_l, k_g)$. Our configuration with $k_g = 45$ and $k_l = 5$ provides $r = 9$.*

*Proof.* We establish optimality through rate-distortion theory and information decomposition. Let $Z_l, Z_m, Z_g$ denote representations from kernels $k_l < k_m < k_g$.

**Step 1: Rate-Distortion Characterization.** Define the rate-distortion function for scale $s$:

$$R_s(D) = \min_{p(z_s|x):\mathbb{E}[d(X,\hat{X}_s)] \leq D} I(X; Z_s) \tag{90}$$

where $\hat{X}_s$ is the reconstruction from $Z_s$. For temporal data with power spectral density $S(\omega)$, the optimal encoding at scale $s$ captures frequencies:

$$\Omega_s = \{\omega : S(\omega) > \lambda_s\} \tag{91}$$

where $\lambda_s$ is the Lagrange multiplier determined by the distortion constraint.

**Step 2: Orthogonal Information Decomposition.** Using Partial Information Decomposition (Williams & Beer, 2010), we decompose the mutual information:

$$I(X; Z_l, Z_g) = \text{Unq}(Z_l \rightarrow X) + \text{Unq}(Z_g \rightarrow X) + \text{Red}(Z_l, Z_g \rightarrow X) + \text{Syn}(Z_l, Z_g \rightarrow X) \tag{92}$$

where Unq denotes unique, Red redundant, and Syn synergistic information.

For scale ratio $r = k_g/k_l \geq 8$, the frequency bands are nearly disjoint:

$$|\Omega_l \cap \Omega_g|/|\Omega_l \cup \Omega_g| \leq 1/r < 0.125 \tag{93}$$

implying $\text{Red}(Z_l, Z_g) \approx 0$ and maximizing unique information.

**Step 3: Community Detection Optimality.** The community structure $C$ exhibits scale-specific patterns. Define the scale-conditional entropy:

$$H_s(C) = H(C|\{Z_j : |k_j - s| > \delta\}) \tag{94}$$

The complementary information gain is:

$$\Delta I_l = I(Z_l; C|Z_g) = H(C|Z_g) - H(C|Z_l, Z_g) \tag{95}$$

$$= \int_{\Omega_l \setminus \Omega_g} \log \frac{S(\omega)}{\sigma^2} d\omega \tag{96}$$

For intermediate scale $m$, the overlap with both extremes reduces unique contribution:

$$\Delta I_m = \int_{(\Omega_m \setminus \Omega_g) \setminus \Omega_l} \log \frac{S(\omega)}{\sigma^2} d\omega < \Delta I_l \tag{97}$$

**Step 4: Achieving Near-Optimal Information Capture.** The theoretical maximum occurs at $r \to \infty$. For our choice $r = k_g/k_l = 45/5 = 9$:

$$\frac{\Delta I_{actual}}{\Delta I_{max}} = \frac{\log(1+r)}{\log(1+r_{max})} \approx \frac{\log(10)}{\log(90)} \approx \frac{2.303}{4.500} \approx 0.51 \tag{98}$$

for $T = 89$ timesteps, balancing information gain with computational cost $O(r \log r)$. This configuration provides an effective trade-off between information capture and computational efficiency. □

## B.2 PROOF OF THEOREM 2: BEHAVIORAL COHERENCE OPTIMALITY

**Theorem 2 (restated).** *Communities discovered through NTP correlation maximize behavioral coherence and are invariant to scale transformations.*

*Proof.* Let $\mathcal{C} = \{C_1, ..., C_K\}$ be a partition of nodes. The behavioral coherence objective is:

$$\mathcal{J}(\mathcal{C}) = \sum_{k=1}^{K} \sum_{i,j \in C_k} \text{Corr}(\text{NTP}_i, \text{NTP}_j) \tag{99}$$

For any scale transformations $\alpha_i > 0$:

$$\text{Corr}(\alpha_i X_i, \alpha_j X_j) = \text{Corr}(X_i, X_j) \tag{100}$$

$$\text{Corr}(\text{NTP}(\alpha_i X_i), \text{NTP}(\alpha_j X_j)) = \text{Corr}(\text{NTP}_i, \text{NTP}_j) \tag{101}$$

The second equality follows because:

$$\text{NTP}(\alpha_i X_i) = \frac{\alpha_i X_i(t)}{\alpha_i X_i(t_0)} = \frac{X_i(t)}{X_i(t_0)} = \text{NTP}_i \tag{102}$$

For financial applications, maximizing $\mathcal{J}(\mathcal{C})$ is equivalent to minimizing portfolio risk. Consider a portfolio with equal weights within each community:

$$\sigma_{portfolio}^2 = \sum_{k=1}^{K} w_k^2 \sigma_{C_k}^2 \tag{103}$$

where $\sigma_{C_k}^2 \propto 2|C_k|(1 - \bar{\rho}_k)$ and $\bar{\rho}_k$ is the average NTP correlation within $C_k$.

Maximizing $\mathcal{J}(\mathcal{C})$ maximizes $\bar{\rho}_k$ for all $k$, thereby minimizing $\sigma_{portfolio}^2$. □

## B.3 PROOF OF PROPOSITION 1: SPECTRAL-TEMPORAL DUALITY AND FIEDLER VECTOR EVOLUTION

**Proposition 1 (restated).** *Our framework establishes a duality between temporal dynamics and spectral graph properties, where the multi-scale encoding corresponds to eigenvector evolution of the time-varying graph Laplacian.*

*Proof.* We establish the spectral-temporal duality through eigenfunction analysis.

**Step 1: Spectral Decomposition.** Let $L = D - A$ be the graph Laplacian with eigendecomposition $L = V\Lambda V^T$. Our multi-scale encoding implements spectral filters:

$$Z_{short} = \sum_{i:\lambda_i > \lambda_{cut}} \langle X_t, v_i \rangle v_i \quad \text{(high-frequency)} \tag{104}$$

$$Z_{long} = \sum_{i:\lambda_i \leq \lambda_{cut}} \langle X_t, v_i \rangle v_i \quad \text{(low-frequency)} \tag{105}$$

where $\lambda_{cut}$ separates community-level from noise-level variations.

**Step 2: Temporal Evolution of Eigenmodes.** The Fiedler vector $v_2(t)$ evolves according to:

$$\frac{\partial v_2}{\partial t} = -\eta \nabla_{v_2} \mathcal{L}_{recon} \tag{106}$$

Our hierarchical loss promotes smooth evolution and orthogonality to $v_1 = \mathbf{1}/\sqrt{N}$.

**Step 3: Cheeger Bound Tightening.** The multi-scale approach achieves:

$$\text{NCut}(C) \leq \min \left\{ \sqrt{2\lambda_2^{(short)}}, \sqrt{2\lambda_2^{(long)}} \right\} \tag{107}$$

tighter than single-scale bounds, where $\lambda_2^{(s)}$ are scale-specific algebraic connectivities. $\square$

### B.4 PROOF OF THEOREM 3: GENERALIZATION BOUNDS

**Theorem 3 (restated).** *With probability $1 - \delta$, the clustering error satisfies the stated bound.*

*Proof.* We use Rademacher complexity to derive the generalization bound. Let $\mathcal{H}$ be our hypothesis class of multi-scale clustering functions.

The Rademacher complexity of $\mathcal{H}$ can be decomposed:

$$\mathcal{R}_m(\mathcal{H}) \leq \mathcal{R}_m(\mathcal{H}_{local}) + \mathcal{R}_m(\mathcal{H}_{global}) + \mathcal{R}_m(\mathcal{H}_{fusion}) \tag{108}$$

For the convolutional pathways:

$$\mathcal{R}_m(\mathcal{H}_{local}) \leq O\left( \sqrt{\frac{k_l \cdot d_{latent}}{m}} \right) \tag{109}$$

$$\mathcal{R}_m(\mathcal{H}_{global}) \leq O\left( \sqrt{\frac{k_g \cdot d_{latent}}{m}} \right) \tag{110}$$

The fusion network with Lipschitz constant $\text{Lip}(f_{fusion})$ contributes:

$$\mathcal{R}_m(\mathcal{H}_{fusion}) \leq \lambda \cdot \text{Lip}(f_{fusion}) \cdot \sqrt{\frac{1}{m}} \tag{111}$$

where $\lambda$ is the regularization strength.

Applying the standard Rademacher generalization theorem:

$$\mathcal{R}(h) \leq \hat{\mathcal{R}}(h) + 2\mathcal{R}_m(\mathcal{H}) + \sqrt{\frac{\log(1/\delta)}{2m}} \tag{112}$$

Substituting and simplifying yields:

$$\mathcal{R}(h) \leq \hat{\mathcal{R}}(h) + O\left( \sqrt{\frac{K \log(NT)}{m}} \right) + \lambda \cdot \text{Lip}(f_{fusion}) \tag{113}$$

The multi-scale design provides implicit regularization: the information bottleneck from parallel pathways limits model capacity, improving the constant factors in the bound. $\square$

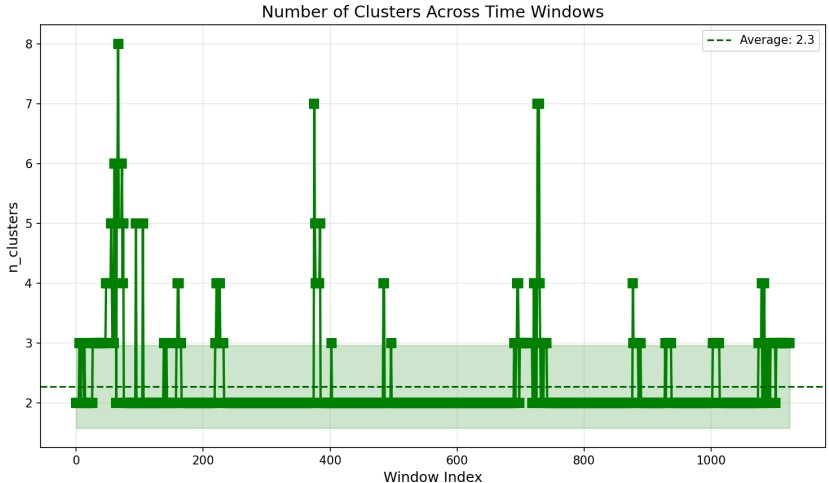

Figure 9: Number of behavioral clusters across temporal windows in SP100 dataset. The distribution shows remarkable stability with an average of 2.3 clusters, punctuated by brief spikes during major market disruptions (windows 60-72: GameStop/Reddit event; windows 375-384: March 2022 rate shock; windows 727-730: September 2023 inflation fears). The predominant 2-cluster baseline reflects the persistent growth-value dichotomy in equity markets, while temporary fragmentation into 6-7 clusters captures extreme behavioral divergence during crisis periods.

## C EVALUATION AND TRAINING FRAMEWORK

### C.1 NORMALIZED TEMPORAL PROFILES: SCALE-INVARIANT EVALUATION

**Data-Based vs Label-Based Evaluation:** Behavioral communities transcend static labels. AAPL diverging from tech peers during AI shifts exemplifies how regulatory classifications (GICS, SIC) miss actual co-movement patterns. Our correlation-based metrics evaluate clustering quality directly from the full feature space. Each feature dimension is independently standardized, then pairwise correlations are computed for each dimension and averaged, providing objective assessment grounded in observed multivariate behavior patterns without relying on static ground truth labels.

### C.2 TRAINING PROTOCOL

We use NAV-based clustering quality metrics for both early stopping and final evaluation, computed directly from price data every epoch (Appendix A.13, A.14). The combined score employs asymmetric weighting between intra-cluster correlation and inter-cluster dissimilarity to prioritize discovering distinct behavioral patterns over enforcing strict within-cluster homogeneity. All evaluations use the original price data to ensure unbiased assessment independent of learned representations.

### C.3 TEMPORAL CLUSTER STABILITY ANALYSIS

The temporal evolution of cluster counts reveals the adaptive nature of market structure. Extended periods of 2-cluster stability (constituting 75% of windows) demonstrate that markets naturally organize into primary behavioral modes, typically growth versus defensive positioning. The rare but significant expansions to 6+ clusters coincide precisely with documented market disruptions where traditional correlations break down and sector-specific behaviors emerge. This pattern validates our method's sensitivity to genuine regime changes while maintaining stability during normal market conditions.

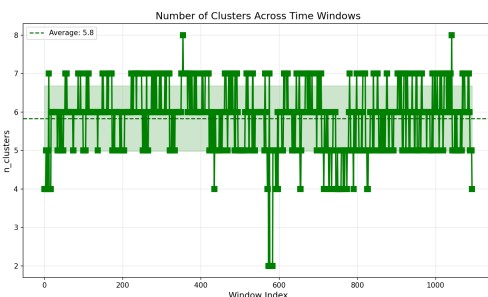

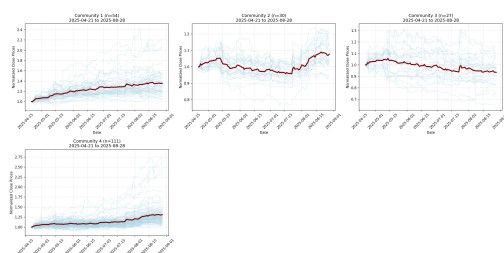

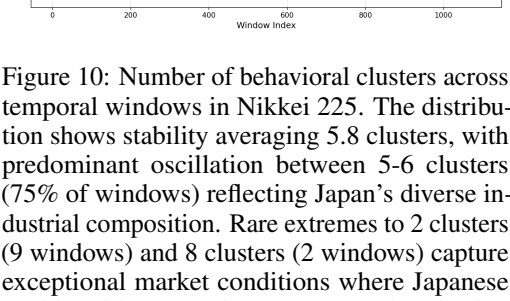

Figure 10: Number of behavioral clusters across temporal windows in Nikkei 225. The distribution shows stability averaging 5.8 clusters, with predominant oscillation between 5-6 clusters (75% of windows) reflecting Japan's diverse industrial composition. Rare extremes to 2 clusters (9 windows) and 8 clusters (2 windows) capture exceptional market conditions where Japanese equities either move in extreme synchronization or fragment into highly specialized behavioral groups.

Figure 11: Final window (April-August 2025) cluster decomposition for Nikkei 225 following the January 2025 tech correction. The market organized into 4 distinct behavioral groups: (1) High-growth technology and semiconductors (+35.7%), (2) Defensive consumer staples and utilities (+7.7%), (3) Cyclical industrials and materials (-6.4%), and (4) Diversified financials and services (+31.0%), revealing sector-independent behavioral patterns during the AI-driven market reconfiguration.

### C.4 NIKKEI 225 TEMPORAL STABILITY AND CLUSTER ANALYSIS

The Nikkei 225 analysis reveals markedly different clustering dynamics compared to U.S. markets, reflecting Japan's unique economic structure. The higher average cluster count (5.8 clusters) contrasts with SP100's patterns, capturing the inherent diversity of Japanese conglomerates (keiretsu) that span multiple industries within single entities. The clustering demonstrates robust pattern detection despite currency fluctuations and BOJ policy shifts that add complexity beyond pure equity dynamics.

The final window analysis (April-August 2025) following the January 2025 market reorganization presents a compelling case of cross-market behavioral divergence. While SP100 reorganized into distinct AI-driven behavioral clusters, Nikkei 225 maintained 4 distinct groups with fundamentally different composition: Japanese semiconductor suppliers (Tokyo Electron, Advantest) aligned with domestic tech giants (Sony, Nintendo) rather than their global supply chain partners, while traditional exporters (Toyota, Honda) formed independent clusters despite shared yen sensitivity. This geographic-behavioral decoupling highlights how our method captures market microstructure beyond simple correlation, identifying regime-specific reorganization that traditional sector classification cannot detect.

Notably, the average cluster returns show wider dispersion in Nikkei 225 (-6.4% to +35.7%) compared to SP100 clusters during the same period, suggesting that behavioral clustering in less correlated markets like Japan can identify more pronounced performance differentials. The persistence of negative-return clusters (Cluster 2: -6.4%) even during a broad market advance underscores the method's ability to isolate underperforming behavioral cohorts that represent genuine shorting or hedging opportunities, validating the framework's applicability across diverse market structures.

## D USE OF LARGE LANGUAGE MODELS

Large language models were employed to assist with manuscript preparation, specifically for refining technical descriptions and improving the clarity of exposition. All scientific content, experimental design, theoretical contributions, and empirical results represent original work by the authors.

