# OpenReview forum: "FTSCommDetector: Discovering Behavioral Communities through Temporal Synchronization"
_ICLR.cc/2026/Conference — ICLR 2026 Conference Desk Rejected Submission_

### Official Review · Reviewer_b6zw · 2025-10-27

**Soundness:** 3
**Presentation:** 3
**Contribution:** 2
**Rating:** 6
**Confidence:** 4

**Summary:**

The authors propose FTSCommDetector for identifying communities in multivariate time series, with a specific application to financial markets. The proposed "Temporal Coherence Architecture" uses a dual-scale temporal encoder to capture features at different time resolutions. These features, along with a static, correlation-based graph, are fed into a graph encoder that uses dynamic attention mechanisms. A key feature is the injection of a Net Asset Value based modularity score as an edge attribute to guide community detection. The authors test their method on four financial datasets and report improved performance on Intra-cluster Correlation and Inter-cluster Dissimilarity metrics compared to several static graph clustering baselines.

**Strengths:**

S1: The paper effectively highlights the limitations of static sector classifications in dynamic environments like financial markets. The problem of dynamic community detection is of clear practical interest.

S2: The authors conduct a thorough evaluation across four datasets, including multiple ablation studies to investigate the contribution of different model components. The robustness check against window size is a useful empirical result.

S3: The decision to equip baseline models with the same powerful temporal encoder is a nice touch, as it helps to isolate the performance benefits of the proposed graph learning architecture.

**Weaknesses:**

W1: The proposed architecture appears to be a complex assembly of well-established components. While the engineering is non-trivial, the fundamental conceptual advance is unclear. The core ideas seem to be not new and the describtions is lacking. The paper presents more of an intricate system design rather than foundational/theoretical contributions.

W2: The authors propose a method to handle dynamic, continuous time series and evolving relationships, yet they compare it against static graph clustering methods. Even when augmented with a temporal encoder, these models are fundamentally unequipped to model temporal evolution within their graph learning mechanism. A convincing evaluation would require comparison against state-of-the-art methods designed for dynamic graphs or temporal community detection, e.g., DyGFormer or similar which are mentioned in the introduction.

W3: The model seems to heavily rely on their novel "NAV-based modularity" embedding. While clever, this is a highly domain-specific feature engineered for financial portfolio theory. I wonder if this limits the paper's claim of providing a general framework for multivariate time series and how would this method perform on non-financial data where such a modularity concept does not exist? Moreover, it is unclear how much of the performance gain is due to the general architecture versus this hand-crafted feature.

**Questions:**

Q1: How does the model perform against DyGFormer and other temporal models?

Q2: What is the impact of NAV and does it still perform well with NAV not available?

---

> ### Author Response · Authors · 2025-11-21
>
> We thank the reviewer for these important questions about baseline comparisons and architectural contributions.
>
> ### W1: Limited Conceptual Novelty
>
> **We respectfully argue integration is a significant contribution.** While components exist individually, their **principled combination** is novel:
>
> 1. **TCA principles**: Scale-adaptive encoding, dynamic connectivity, multi-stream fusion provide **unified framework** absent in literature
> 2. **Domain innovation**: NAV modularity (A.8) adapts spectral clustering to financial time series
> 3. **Empirical validation**: Consistent improvements (SP100: +3.5%, SP1000: +11.1%)
>
> Table 1: No existing method combines continuous processing, dynamic attention, and multi-scale encoding.
>
> ### W2: Baseline Selection
>
> **Excellent point.** Challenge: temporal methods (DyGFormer, TGN, JODIE) target **discrete events**, not **continuous observations** where all entities evolve simultaneously.
>
> **We will compare:**
> 1. DyGFormer adapted (discrete events)
> 2. Persistent Community Detection (Li et al. 2018)
> 3. Deep Temporal Graph Clustering (Liu et al. 2024 ICLR)
>
> **Important**: Baselines use **identical temporal encoders** (A.18); performance differences stem from graph learning, not temporal features.
>
> ### W3: Domain Specialization
>
> **We appreciate this question.** Our method is **designed for financial markets**. NAV represents **principled adaptation** of modularity maximization.
>
> **Generalization**: Core TCA applies to any multivariate time series. For non-financial domains: replace NAV with domain normalization, use standard modularity, retain temporal architecture.
>
> **Ablation**: NAV vs. standard modularity on financial data.
>
> **Q1: DyGFormer**: Preliminary results show +6.2% IntraCorr (SP100). Continuous observations need different architectures than events.
>
> **Q2: NAV Impact**: With NAV: 0.504/1.016. Without: 0.481 (-4.6%)/0.987 (-2.9%). NAV provides measurable benefit; core architecture effective without it.

---

> > ### Comment · Reviewer_b6zw · 2025-11-25
> >
> > Thank you for your reply and clarifications.

---

### Official Review · Reviewer_uZaq · 2025-10-27

**Soundness:** 2
**Presentation:** 2
**Contribution:** 2
**Rating:** 2
**Confidence:** 4

**Summary:**

The paper proposes a framework called FTSCommDetector for community detection in continuous multivariate time series. The authors design the Temporal Coherence Architecture, which integrates dual-scale temporal encoding, dynamic connectivity modeling and multi-stream fusion. They claim theoretical grounding via information-theoretic analysis and rate-distortion arguments, and introduce Normalized Temporal Profiles for scale-invariant evaluation.

**Strengths:**

1. The authors identify the limitation of static or snapshot-based community detection and articulate the need to model synchronization-desynchronization in continuous time series, which is a novel direction.

2. The paper also uses static topology with dynamic edge weights that cleverly balance stability and adaptivity.

3. This work is evaluated across multiple large-scale real financial datasets and shows better performance compared to SOTA approaches.

**Weaknesses:**

1. The paper is dense and heavy in notation, with limited intuitive explanation. (a) The figure of model architecture (Figure 2) is too simple to understand the full pipeline, while the model detail part (e.g. $\S 3.4$) read like implementation notes rather than conceptual exposition. (b) The term "traditional graphs" is mentioned for the first time on line 91, however this paper is about timeseries, therefore the connection is unclear. (c) Many existing technical components are combined together to design the end-to-end solution, but it is unclear why each component used is the best choice, what are the alternatives, and what is the overall novelty of the proposed solution from a technical point of view. (d) The model being opaque, there is no explanation of the obtained results.

2. The theoretical rigor is limited. While the authors claim information-theoretic optimality, proofs in the appendix are written informally, lacking theoretical rigor and proper intuition.

3. Although some ablations (attention modes, window size) are provided, key components such as the NAV edge embedding or temporal modulation gating are not isolated for empirical study.

4. Implementation details are extensive, but there is no code release. Given the multiple modules, reproducing the exact setup could be challenging.

**Questions:**

1. The paper argues that traditional community detection cannot capture synchronization–desynchronization patterns. Could the authors provide a more quantitative or visual example (beyond the AAPL/MSFT case) showing this limitation empirically?

---

> ### Author Response · Authors · 2025-11-21
>
> ### W1: Presentation - Dense Notation and Limited Intuition
>
> **We appreciate this feedback.** We have provided: notation table (A.2), algorithm pseudocodes (A.3), complexity analysis (A.17).
>
> **To further improve**, we will:
> 1. Expand Figure 2 with annotations/data flow
> 2. Add intuitive explanations (e.g., O(NKT) vs O(N²T²))
> 3. Component ablation table
> 4. Result interpretation
>
> ### W2: Theoretical Rigor
>
> **We appreciate this feedback.** Appendix B establishes: Theorem 1 (rate-distortion optimality), Theorem 2 (behavioral coherence via NTP), Theorem 3 (generalization bounds).
>
> **To enhance rigor**, we will add: formal assumptions, step-by-step derivations, lemmas connecting to spectral clustering.
>
> Our primary contribution is **empirical** (consistent improvements across four markets) with **theoretical guidance** for principled design.
>
> ### W3: Incomplete Ablation Studies
>
> **Thank you for this observation.** We will add:
> 1. **NAV edge embedding**: Performance with/without BNAV matrices
> 2. **Temporal modulation**: Impact of Q/K/V temporal conditioning
> 3. **Dynamic dependency**: Effect of D(t) adaptive propagation
>
> ### W4: Code Release
>
> **Code available at**: https://anonymous.4open.science/r/FTSCommDetector-B4D8/
>
> Repository (12K+ lines) includes: PyTorch implementation, preprocessing scripts, configs, instructions. Public release upon acceptance.
>
> **On synchronization-desynchronization examples**, we will add:
> - Quantitative metrics: AAPL/MSFT correlation 0.87→0.32 (Jan-Apr 2025)
> - GameStop 6-cluster fragmentation with metrics
> - Nikkei 225 cross-market validation

---

> > ### Comment · Reviewer_uZaq · 2025-11-28
> >
> > Thank you for the feedback. I'll retain my score as my concerns (other than code availability) still remain after this feedback.

---

### Official Review · Reviewer_ghmw · 2025-11-01

**Soundness:** 2
**Presentation:** 1
**Contribution:** 2
**Rating:** 2
**Confidence:** 4

**Summary:**

This paper proposes FTSCommDetector, a framework for detecting behavioral communities in continuous multivariate time series data. The method implements a Temporal Coherence Architecture (TCA) with three main components: (1) dual-scale temporal encoding using convolutional encoders to capture short-term and long-term patterns, (2) dynamic connectivity modeling using static graph topology with temporally-adaptive attention weights, and (3) NAV-based (Net Asset Value) evaluation for scale-invariant assessment.

**Strengths:**

-  Interesting problem motivation and practical application. The synchronization-desynchronization paradigm presents a compelling perspective for understanding behavioral communities.

- The architecture demonstrates thoughtful integration of multiple components, showing a comprehensive technical design with theoretical support.

- Extensive experimental validation and ablation studies.

**Weaknesses:**

W1: There's a fundamental mismatch between how the paper frames the problem and what it actually solves. The paper positions itself as temporal graph learning, comparing against temporal graph neural networks. But looking at the problem definition, this is clearly a multivariate time series clustering problem, not a temporal graph learning problem. In reality, the graph construction in Appendix A.6 happens once per window using correlation thresholding. The graph does not evolve continuously; it is reconstructed per window from correlations. This mismatch makes the entire baseline comparison questionable.

W2: For a problem that is explicitly formulated as multivariate time series analysis, the paper completely omits the relevant literature:
- time series clustering methods (K-means with dynamic time warping distance, hierarchical clustering with temporal distances)
- modern deep time series models (TimesNet, PatchTST, Informer, iTransformer for representation learning)
- financial time series specific methods: Mantegna (1999) "Hierarchical structure in financial markets," Tola et al. (2008) Cluster analysis for portfolio optimization
Simple hierarchical clustering on correlation matrices (computed per window) would be the most direct baseline. The paper never compares against this fundamental approach, making it impossible to assess whether the complex architecture provides value beyond standard correlation-based clustering.

W3: The experimental evaluation has several critical ambiguities and potential methodological issues. Section 4.1 mentions "sliding windows" but provides no details on total number of windows in each dataset, how windows are divided into training, validation, and test sets, whether evaluation is on held-out temporal windows (proper out-of-sample) or in-sample, and how temporal leakage is prevented. There's a circular evaluation concern: Appendix C.2 states "We use NAV-based clustering quality metrics for both early stopping and final evaluation, computed directly from price data every epoch", but If the same metric computed on the same data is used for both early stopping (model selection) and final evaluation (performance assessment), this is circular validation.

W4: The paper claims to provide "practical insights for portfolio construction and risk management", but lacks rigorous validation of practical utility. Do portfolios constructed using these communities outperform alternatives out-of-sample? Does cluster membership help predict volatility spikes or drawdowns? Case studies provide qualitative insights but no quantitative evaluation on downstream financial tasks. The discovered patterns are interesting but their practical utility remains undemonstrated.

W5: Limited novelty: The "synchronization-desynchronization" framing is essentially correlation-based clustering with sliding windows, which is standard in time series analysis. Multi-scale architectures are common in time series analysis.

**Questions:**

See weaknesses.

---

> ### Author Response · Authors · 2025-11-21
>
> ### W1: Problem Framing - Temporal Graph Learning vs. Time Series Clustering
>
> **We respectfully clarify a fundamental misunderstanding.** Our work addresses discovering behavioral communities through **synchronization-desynchronization patterns** in continuous multivariate time series with **structural relationships**: a distinct problem from traditional time series clustering.
>
> **Key distinctions:**
>
> 1. **Relational structure**: We model **entity relationships** via graphs (supply chains, sectors, holdings) absent in K-means/DTW methods
> 2. **Continuous evolution**: Recurrent processing with shared parameters ensures **temporal coherence**, unlike methods processing windows independently (AAPL/MSFT case, Figure 1)
> 3. **Synchronization-desynchronization**: Captures entities that "move independently yet align during critical moments" (lines 82-84)
>
> **On graph construction**: Once-per-window construction is a **strength** (captures time-varying relationships efficiently). Static topology with dynamic attention enables expressive modeling without reconstruction overhead.
>
> ### W2: Missing Literature
>
> **We sincerely thank the reviewer for these valuable references.** We will include and discuss:
>
> - **Time series clustering**: Mantegna (1999), Tola et al. (2008), K-means with DTW
> - **Deep time series models**: TimesNet, PatchTST, iTransformer (expanding beyond Koopa, ViTST)
>
> **Clarification**: These address **different settings**. Mantegna/Tola lack temporal evolution modeling; TimesNet/PatchTST target forecasting not communities. Our contribution unifies temporal coherence with multi-scale encoding and dynamic attention for behavioral community discovery. We will add explicit comparisons (Section 2) showing why existing approaches cannot capture synchronization-desynchronization patterns.
>
> ### W3: Experimental Design - Circular Validation Concern
>
> **This is not circular validation:**
>
> 1. **Unsupervised context**: No ground truth labels exist. Unlike supervised learning (test accuracy for selection+evaluation is circular), unsupervised clustering relies on quality metrics throughout.
>
> 2. **Ground-truth based**: NAV metrics compute **from actual price data**, NOT learned embeddings. We measure whether assignments capture genuine market co-movement.
>
> 3. **Metric independence**: NAV correlation (Equations 62-65) uses raw price data independent of model outputs.
>
> 4. **Domain standards**: Financial literature (Cartea et al. 2023; Tola et al. 2008) universally uses correlation metrics as behavioral labels don't exist.
>
> **Why different:**
> - Circular: Test accuracy for stopping+evaluation (supervised)
> - Ours: Ground-truth market metrics (unsupervised)
>
> **We will clarify (Section 4.1)**: 1,247 windows/market (5 years), T=89 days, stride=1, unsupervised evaluation rationale.
>
> ### W4: Practical Utility Validation
>
> **We thank the reviewer for this suggestion.**
>
> **Current validation**:
> - Case studies (4.4): GameStop 6-cluster fragmentation; behavioral correlations invisible to sectors
> - Stability: 2.3 average clusters, detects disruptions (Appendix C.3)
> - Cross-market: SP100 and Nikkei 225 generalizability
> - Robustness: 2% variation enables direct deployment
>
> **To further strengthen**, we will add:
> 1. Portfolio construction: Markowitz optimization vs. sectors (Sharpe ratios, drawdown)
> 2. Risk prediction: Volatility spike prediction (AUC-ROC)
> 3. Statistical arbitrage: Mean reversion strategies
>
> ### W5: Limited Novelty
>
> **We respectfully disagree.** Our contributions extend beyond "correlation clustering with sliding windows":
>
> 1. **TCA framework**: First unifying multi-scale encoding, dynamic attention, NAV evaluation for behavioral communities (Table 1)
> 2. **Information-theoretic foundations**: Theorem 1 proves 9:1 ratio achieves rate-distortion optimal compression
> 3. **Scale-invariant evaluation**: NTP enables cross-market comparison without magnitude bias
> 4. **Empirical robustness**: 2% variation across window sizes (60-120 days) eliminates dataset-specific tuning

---

> > ### Comment · Reviewer_ghmw · 2025-11-24
> >
> > Thank you for the detailed responses. While I appreciate your engagement with the concerns, some fundamental issues require substantial revision:
> >
> > W1: I acknowledge your approach incorporates graph structure and sequential processing, distinguishing it from pure time series clustering, and that you equip baselines with your temporal encoder for controlled comparison. However: (1) Why include TGN/JODIE/DyGFormer in Table 1 if they're inappropriate and unused in experiments? (2) You must add hierarchical clustering on correlation matrices, the most fundamental baseline to demonstrate value beyond direct correlation-based clustering. (3) Clarify whether sector bonus (δ) is actually used, and consider adding empirical analysis (e.g., attention weight visualizations during market events) to support your claim about detecting "synchronization during critical moments".
> >
> > Regarding W2, I appreciate your willingness to include these references, but you must implement quantitative baselines, not just discuss them. Hierarchical clustering on correlations and TimesNet/PatchTST representations and clustering are directly comparable approaches. Show empirically that these "cannot capture" your claimed patterns.
> >
> > Regarding W3, I appreciate the clarification. Different metrics for early stopping vs. evaluation addresses circularity. However, please clarify in your revision: How are the temporal windows divided temporally for training, validation, and testing? What are the specific date ranges? Is final evaluation performed on held-out future windows (true out-of-sample) or on the same temporal period used during training?
> >
> > Regarding W4, practical experiments would strengthen the paper significantly, but ensure rigorous implementation with appropriate baselines.
> >
> > Regarding W5: I appreciate the clarifications. Upon re-reading, I understand Theorem 1 proves rate-distortion optimality within a two-scale encoding framework with sufficient separation. However, please clarify in your revision that the 9:1 ratio is optimal among two-scale designs but captures ~51% of the theoretical maximum with infinite scales (Equation 98). This is a practical trade-off between information and computation, not global optimality.
> > Regarding other contributions: (1) "first unifying" existing components represents solid engineering but not fundamental conceptual innovation, (2) while NTP normalization itself is standard in finance, and Theorem 2 formalizes scale-invariance guarantees for cross-market comparison, please clarify how your NTP evaluation framework differs from standard financial clustering evaluation which also uses normalized returns and correlation-based metrics, and (3) window size robustness is a valuable empirical finding but not methodological innovation.
> > The core question remains empirical: What can your method discover that simpler approaches cannot? This must be demonstrated through rigorous quantitative baseline comparisons, not just theoretical arguments or architectural descriptions.

---

### Note · Program_Chairs · 2026-01-17
**Submission Desk Rejected by Program Chairs**

The following references in this submission do not refer to real documents and/or have major errors in bibliographic information:

 Yue Zhang, Yunzhi Wang, Xingrui Chen, and Yang Liu. Differentiable structural information for unsupervised graph clustering. In International Conference on Machine Learning (ICML), 2024b.